# Scalable Decentralized Learning with Teleportation

**Yuki Takezawa**[*]
Kyoto University, OIST

**Sebastian U. Stich**
CISPA Helmholtz Center for Information Security

## Abstract

Decentralized SGD can run with low communication costs, but its sparse communication characteristics deteriorate the convergence rate, especially when the number of nodes is large. In decentralized learning settings, communication is assumed to occur on only a given topology, while in many practical cases, the topology merely represents a preferred communication pattern, and connecting to arbitrary nodes is still possible. Previous studies have tried to alleviate the convergence rate degradation in these cases by designing topologies with large spectral gaps. However, the degradation is still significant when the number of nodes is substantial. In this work, we propose Teleportation. Teleportation activates only a subset of nodes, and the active nodes fetch the parameters from previous active nodes. Then, the active nodes update their parameters by SGD and perform gossip averaging on a relatively small topology comprising only the active nodes. We show that by activating only a proper number of nodes, Teleportation can completely alleviate the convergence rate degradation. Furthermore, we propose an efficient hyperparameter-tuning method to search for the appropriate number of nodes to be activated. Experimentally, we showed that Teleportation can train neural networks more stably and achieve higher accuracy than Decentralized SGD.

## 1 Introduction

Distributed learning has emerged as an important paradigm for privacy preservation and large-scale machine learning. In centralized approaches, such as federated learning (McMahan et al., 2017; Kairouz et al., 2019) and All-Reduce, nodes update their parameters by using their local dataset and then compute the average parameters of these nodes. Since computing the average of many nodes incurs huge communication costs, communication is the major bottleneck in the training time. In reducing communication costs, decentralized learning has attracted significant attention (Lian et al., 2017; 2018; Assran et al., 2019; Koloskova et al., 2020b). In decentralized learning, a node exchanges parameters with its few neighboring nodes and then computes their weighted average to approximate the average of all nodes. This procedure is called gossip averaging. Since each node only needs to communicate with a few neighboring nodes, decentralized learning is more communication efficient than centralized counterparts (Lian et al., 2017; Assran et al., 2019; Ying et al., 2021).

While decentralized learning can run with low communication costs, the degradation of the convergence rate due to its sparse communication characteristics is a trade-off (Koloskova et al., 2020b). Especially, when the number of nodes is large, the parameters held by each node are likely to be far away during the training, and gossip averaging deteriorates the convergence rate. This makes it challenging for decentralized learning to scale to a large number of nodes.

In decentralized learning literature, it is commonly assumed that communication can only occur on a given topology. However, in many practical cases, the topology merely represents a preferred communication pattern, and connecting to arbitrary nodes is still possible, e.g., in a data center or the case where nodes are connected on the Internet. Since we can choose which topology to use in these cases, many prior studies designed topologies to reconcile the communication efficiency and convergence rate, attempting to alleviate the degradation of the convergence rate caused by a large number of nodes (Wang et al., 2019; Ying et al., 2021; Ding et al., 2023; Takezawa et al., 2023b).

---

[*]This work was done while YT visited the CISPA Helmholtz Center for Information Security.

Specifically, the communication costs are determined by the maximum degree of the topology, and the convergence rate is determined by its spectral gap (Wang et al., 2019; Koloskova et al., 2020b). Thus, the prior studies designed topologies with large spectral gaps and small maximum degrees and proposed performing gossip averaging on these topologies. However, the approximate average estimated by gossip averaging deviates from the exact average of all nodes as the number of nodes increases, and the convergence rate remains to degrade when the number of nodes is substantial.

In this study, we ask the following question: *Can we develop a decentralized learning method whose convergence rate does not degrade as the number of nodes increases?* Our work yielded an affirmative answer with the proposal of TELEPORTATION. TELEPORTATION can completely eliminate the degradation of the convergence rate caused by increasing the number of nodes, and the convergence rate is consistently improved as the number of nodes increases. Specifically, TELEPORTATION activates only a subset of nodes and initializes their parameters with the parameters of previous active nodes. Then, the active node updates its parameters by SGD and performs gossip averaging on a relatively small topology comprising only the active nodes. By activating only a proper number of nodes, gossip averaging can make the parameters of each node reach the consensus fast, and the parameters of each node can be prevented from being far away even when the total number of nodes is large. We show that by activating an appropriate number of nodes, the degradation of the convergence rate can be completely alleviated. Furthermore, we propose an efficient hyperparameter-tuning method to search for the proper number of active nodes. For an arbitrary number of nodes, the proposed hyperparameter-tuning method can find the appropriate number of nodes to be activated within $2T$ iterations in total, where $T$ is the number of iterations to run TELEPORTATION once. Experimentally, we demonstrated that TELEPORTATION can converge faster than Decentralized SGD and train neural networks more stably.

**Note:** In this work, we consider a setting where any two nodes can exchange parameters as required, similar to previous studies on topology design for decentralized learning (Marfoq et al., 2020; Ying et al., 2021; Song et al., 2022; Takezawa et al., 2023b; Ding et al., 2023). A data center is an example, and Assran et al. (2019) demonstrated that decentralized methods can train neural networks faster than All-Reduce. In addition to a data center, when nodes are connected to the Internet, any two nodes can communicate by specifying the IP address. TELEPORTATION is not applicable in the presence of a pair of nodes that cannot communicate, as in wireless sensor networks.

**Notation:** A graph $\mathcal{G}$ is represented by $(V, E)$ where $V$ is a set of nodes and $E$ is a set of edges. For simplicity, we also write $(\{1, \cdots, n\}, E)$ to represent a graph $\mathcal{G}$ where $n$ is the number of nodes. $\boldsymbol{I}_d$ denotes the $d$-dimentional identify matrix, $\mathbf{0}$ denotes a vector with all zeros, and $\mathbf{1}$ denotes a vector with all ones.

## 2 PRELIMINARY

**Problem Setting:** We consider the following problem where the loss functions are distributed among $n$ nodes:

$$\min_{\boldsymbol{x} \in \mathbb{R}^d} \left[ f(\boldsymbol{x}) \coloneqq \frac{1}{n} \sum_{i=1}^{n} f_i(\boldsymbol{x}) \right], \quad f_i(\boldsymbol{x}) \coloneqq \mathbb{E}_{\xi_i \sim \mathcal{D}_i} \left[ F_i(\boldsymbol{x}; \xi_i) \right], \tag{1}$$

where $\boldsymbol{x}$ is a model parameter, $\mathcal{D}_i$ is a training dataset that the $i$-th node has, $\xi_i$ is a date sample following $\mathcal{D}_i$, and the local loss function $f_i : \mathbb{R}^d \to \mathbb{R}$ is defined as the expectation of $F_i(\cdot; \xi_i)$ over data sample $\xi_i$. Following previous works (Koloskova et al., 2020a; Yuan et al., 2021; Lin et al., 2021), we assume that the loss functions satisfy the following assumptions.

**Assumption 1.** *There exists $f^\star > -\infty$ that satisfies $f(\boldsymbol{x}) \geq f^\star$ for any $\boldsymbol{x} \in \mathbb{R}^d$.*

**Assumption 2.** *There exists $L \geq 0$ that satisfies for any $\boldsymbol{x}, \boldsymbol{y} \in \mathbb{R}^d$ and $i \in \{1, 2, \cdots, n\}$,*

$$\|\nabla f_i(\boldsymbol{x}) - \nabla f_i(\boldsymbol{y})\| \leq L \|\boldsymbol{x} - \boldsymbol{y}\|. \tag{2}$$

**Assumption 3.** *There exists $\sigma \geq 0$ that satisfies for any $\boldsymbol{x} \in \mathbb{R}^d$ and $i \in \{1, 2, \cdots, n\}$,*

$$\mathbb{E}\|\nabla F_i(\boldsymbol{x}; \xi_i) - \nabla f_i(\boldsymbol{x})\|^2 \leq \sigma^2. \tag{3}$$

**Assumption 4.** *There exists $\zeta \geq 0$ that satisfies for any $\boldsymbol{x} \in \mathbb{R}^d$,*

$$\frac{1}{n} \sum_{i=1}^{n} \|\nabla f_i(\boldsymbol{x}) - \nabla f(\boldsymbol{x})\|^2 \leq \zeta^2. \tag{4}$$

**Decentralized SGD:** One of the most fundamental decentralized learning methods to solve Eq. (1) is Decentralized SGD (Lian et al., 2017). Let $\boldsymbol{x}_i \in \mathbb{R}^d$ denote the $i$-th node parameter. Once the topology comprising $n$ nodes $\mathcal{G}_n = (\{1, \cdots, n\}, E)$ is specified, Decentralized SGD updates $\boldsymbol{x}_i$ as follows:

$$\boldsymbol{x}_i^{(t+1)} = \sum_{j=1}^{n} W_{ij} \left( \boldsymbol{x}_j^{(t)} - \eta \nabla F_j(\boldsymbol{x}_j^{(t)}; \xi_j^{(t)}) \right),$$

where $\eta > 0$ is the step size and $W_{ij} \in [0, 1]$ is the weight of edge $(i, j) \in E$ that satisfies $\sum_{i=1}^{n} W_{ij} = \sum_{j=1}^{n} W_{ij} = 1$. The following assumption is commonly used to represent how well the topology $\mathcal{G}_n$ is connected (Yuan et al., 2021; Koloskova et al., 2020b; 2021; Aketi et al., 2023).

**Assumption 5.** *Let $\lambda_i$ is the $i$-th largest eigenvalue of $\boldsymbol{W} \in [0, 1]^{n \times n}$. $\boldsymbol{W}$ is a doubly stochastic matrix, and $p_n := 1 - \max\{|\lambda_2|, |\lambda_n|\}^2$ satisfies $p_n \in (0, 1]$. That is, it holds that*

$$\left\| \boldsymbol{X}\boldsymbol{W} - \bar{\boldsymbol{X}} \right\|_F^2 \leq (1 - p_n) \left\| \boldsymbol{X} - \bar{\boldsymbol{X}} \right\|_F^2,$$

*for any $\boldsymbol{X} \in \mathbb{R}^{d \times n}$ where $\bar{\boldsymbol{X}} := \frac{1}{n} \boldsymbol{X} \mathbf{1} \mathbf{1}^\top$.*

We write $p_n$ to emphasize that $p_n$ depends on $n$. A small $p_n$ indicates that the topology is poorly connected, and a large $p_n$ indicates that the topology is well connected. Generally, $p_n$ approaches zero as the number of node $n$ increases. For instance, when a ring is used as the underlying topology, $p_n$ is $\Omega(n^{-2})$, which rapidly decreases as $n$ increases (Nedić et al., 2018). Under these assumptions, Decentralized SGD converges with the following rate.

**Proposition 1** (Koloskova et al. (2020b)). *Suppose that Assumptions 1, 2, 3, 4, and 5 hold. Let $\{\{\boldsymbol{x}_i^{(t)}\}_{i=1}^n\}_{t=0}^T$ denote the parameters generated by Decentralized SGD. Then, there exists the step size $\eta$ that satisfies:*

$$\frac{1}{T+1} \sum_{t=0}^{T} \mathbb{E}\|\nabla f(\bar{\boldsymbol{x}}^{(t)})\|^2 \leq \mathcal{O}\left( \sqrt{\frac{Lr_0\sigma^2}{nT}} + \left( \frac{L^2 r_0^2 (p_n\sigma^2 + \zeta^2)(1 - p_n)}{T^2 p_n^2} \right)^{\frac{1}{3}} + \frac{Lr_0}{Tp_n} \right), \tag{5}$$

*where $r_0 := f(\bar{\boldsymbol{x}}^{(0)}) - f^\star$ and $\bar{\boldsymbol{x}}^{(t)} := \frac{1}{n} \sum_{i=1}^{n} \boldsymbol{x}_i^{(t)}$.*

As the number of nodes $n$ increases, the first term $\mathcal{O}(\sqrt{\frac{Lr_0\sigma^2}{nT}})$ is improved, while the second and third terms degrade since $p_n$ reaches zero. Thus, as the number of nodes $n$ increases substantially, the second and third terms dominate the convergence rate, and the convergence rate deteriorates. In many practical cases, $\mathcal{G}_n$ merely represents a preferred communication pattern, and any two nodes can communicate as needed, as introduced in Sec. 1. Since we can choose which topology to use in these cases, many recent studies have attempted to alleviate the convergence rate degradation by proposing topologies with large $p_n$ (Ying et al., 2021; Ding et al., 2023; Takezawa et al., 2023b). However, their $p_n$ still approach zero as $n$ increases, and the convergence rate degrades when $n$ is substantially large. Table 2 in Sec. D lists the convergence rates of Decentralized SGD with various topologies.

## 3 PROPOSED METHOD

In this section, we propose TELEPORTATION, which can completely eliminate the degradation of the convergence rate caused by increasing the number of nodes. In the remainder of this section, we describe TELEPORTATION in Sec. 3.1 and analyze its convergence rate in Sec. 3.2. Then, we propose an efficient hyperparameter search method in Sec. 3.3 and analyze its convergence rate in Sec. 3.4.

### 3.1 TELEPORTATION

$p_n$ represents the speed at which the gossip averaging makes the parameter of each node $\{\boldsymbol{x}_i\}_{i=1}^n$ reach the average $\frac{1}{n} \sum_{i=1}^{n} \boldsymbol{x}_i$. As the number of nodes $n$ increases, averaging all node parameters

---

**Algorithm 1** Simple version of TELEPORTATION

---

1: **Input:** the number of nodes $n$, set of nodes $V_n$, number of active nodes $k$, total number of iteration $T$, step size $\eta$, and topology comprising $k$ active nodes $\mathcal{G}_k = (\{1, \cdots, k\}, E)$.
2: **for** $t \in \{0, 1, \cdots, T\}$ **do**
3:      sample next active $k$ nodes $V_{\text{active}}^{(t)}$ from $V_n$ without replacement, and assign $\{1, 2, \cdots, k\}$ to variables $\{\texttt{token\_id}_i^{(t)} \mid v_i \in V_{\text{active}}^{(t)}\}$ randomly without overlap.
4:      **for** $i \in \{1, 2, \cdots, n\}$ **in parallel do**
5:          **if** $v_i \in V_{\text{active}}^{(t)}$ **then**[a]
6:              **if** $t > 0$ **then**
7:                  Let $v_j \in V_{\text{active}}^{(t-1)}$ denote the node such that $\texttt{token\_id}_j^{(t-1)} = \texttt{token\_id}_i^{(t)}$.
8:                  receive $\boldsymbol{x}_j^{(t)}$ from $v_j \in V_{\text{active}}^{(t-1)}$, and $\boldsymbol{x}_i^{(t)} \leftarrow \boldsymbol{x}_j^{(t)}$.
9:              **end if**
10:              $\boldsymbol{y}_i^{(t)} \leftarrow \boldsymbol{x}_i^{(t)} - \eta \nabla F_i(\boldsymbol{x}_i^{(t)}; \xi_i^{(t)})$.
11:              receive $\boldsymbol{y}_j^{(t)}$ from $v_j \in \{v_j \in V_{\text{active}}^{(t)} \mid (\texttt{token\_id}_i^{(t)}, \texttt{token\_id}_j^{(t)}) \in E\}$.
12:              $\boldsymbol{x}_i^{(t+1)} \leftarrow \sum_{v_j \in V_{\text{active}}^{(t)}} W_{\texttt{token\_id}_i^{(t)}, \texttt{token\_id}_j^{(t)}} \boldsymbol{y}_j^{(t)}$.
13:          **end if**
14:      **end for**
15: **end for**

---

[a]The update rule of the parameters of the inactive nodes is not described because the parameters held by the inactive nodes are discarded and initialized with the parameters of the other active nodes when they are activated in line 8. The parameters of inactive nodes do not affect the behavior of active node parameters.

---

becomes difficult with gossip averaging, and $p_n$ reaches zero. Thus, as $n$ increases, the parameters of each node $\{\boldsymbol{x}_i\}_{i=1}^n$ comes to be apart during the training, and the convergence rate deteriorates, as discussed in Sec. 2.

To prevent the parameters in each node $\{\boldsymbol{x}_i\}_{i=1}^n$ from being away when the number of nodes $n$ is large, we propose updating the parameters of only a small subset of nodes and performing gossip averaging on a small topology. We refer to the proposed method as **TELEPORTATION**.[1] Let $k \in \{1, \cdots, n\}$ denote the number of active nodes and $\mathcal{G}_k = (\{1, \cdots, k\}, E)$ be the topology comprising $k$ active nodes. Essentially, TELEPORTATION consists of the following three steps:

(1) We sample $k$ active nodes $V_{\text{active}}^{(t)}$ from $V_n$ without replacement and assign $\{1, 2, \cdots, k\}$ to variables $\{\texttt{token\_id}_i^{(t)} \mid v_i \in V_{\text{active}}^{(t)}\}$ without overlap.[2]

(2) The active node $v_i \in V_{\text{active}}^{(t)}$ fetches the parameters from the previous active node $v_j \in V_{\text{active}}^{(t-1)}$ whose $\texttt{token\_id}_j^{(t-1)}$ stores the same value as that of $\texttt{token\_id}_i^{(t)}$. Then, the active node $v_i$ initializes its parameters with the fetched parameter and updates them by SGD.

(3) The active node $v_i \in V_{\text{active}}^{(t)}$ exchanges its parameters with its neighbors $\{v_j \in V_{\text{active}}^{(t)} \mid (\texttt{token\_id}_i^{(t)}, \texttt{token\_id}_j^{(t)}) \in E\}$ and computes the weighted average with them.

We show the pseudo-code and illustration in Alg. 1 and Fig. 1. In the above implementation, the first step does not start until the third step is completed. We can solve this issue so that gossip averaging in the third step is performed on the next active node $v_j$, and the first and third steps can be executed simultaneously. We show the optimized version in Sec. A.

In TELEPORTATION, gossip averaging is performed on a relatively small topology $\mathcal{G}_k$, which can make the parameters of $k$ active nodes reach the average fast. Therefore, using the proper number of active nodes $k$, we can mitigate the degradation of the convergence rate by increasing the number of nodes $n$. In the next section, we analyze the convergence rate and show that our proposed method successfully circumvents the negative effect of increasing the number of nodes $n$.

---

[1]We named our proposed method TELEPORTATION to reflect the manner in which the active nodes copy their parameters to the next active nodes.

[2]The first step can be performed in a decentralized manner by sharing the same seed value to sample active nodes before starting the training.

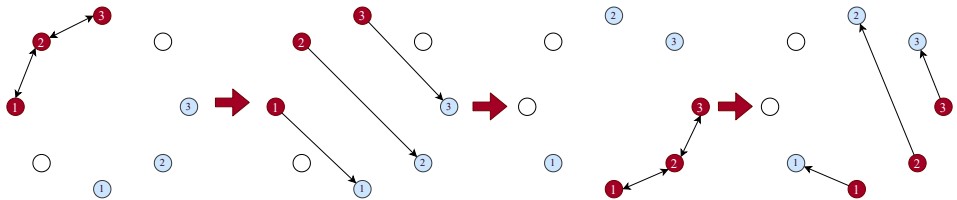

Figure 1: Illustration of Alg. 1 with $n = 8$ and $k = 3$. We use a line as the topology consisting of active nodes $\mathcal{G}_k = (\{1, 2, 3\}, \{(1, 1), (1, 2), (2, 2), (2, 3), (3, 3)\})$. The black nodes represent active nodes, and the number written on the node is $\texttt{token\_id}_i^{(t)}$. The blue nodes represent the next active nodes, and the number on the node is $\texttt{token\_id}_i^{(t+1)}$. The first and third graphs from the left represent the communication in line 12, and the other graphs represent the communication in line 8.

**Comparison with Client Sampling:** TELEPORTATION randomly samples a subset of nodes and activates them, but TELEPORTATION is different from the client sampling (Liu et al., 2022). TELEPORTATION can prevent the parameters of nodes from being away by using the small $k$ because gossip averaging is performed on the small topology $\mathcal{G}_k$. In contrast to TELEPORTATION, client sampling incites the parameters of nodes to drift away because gossip averaging is performed on the topology comprising all nodes, and nodes can exchange parameters only when two neighboring nodes are sampled simultaneously, which makes it more difficult for gossip averaging to make the parameters reach the consensus. Thus, client sampling cannot alleviate the degradation of the convergence rate caused by a large number of nodes $n$ and degrades the convergence rate as well as the vanilla Decentralized SGD (see Sec. E for more detailed discussion).

### 3.2 CONVERGENCE ANALYSIS OF TELEPORTATION

We assume that the topology comprising $k$ active nodes $\mathcal{G}_k$ satisfies the following assumption.

**Assumption 6.** *Let $\lambda_i$ is the $i$-th largest eigenvalue of $\boldsymbol{W} \in [0, 1]^{k \times k}$. $\boldsymbol{W}$ is a doubly stochastic matrix, and $p_k := 1 - \max\{|\lambda_2|, |\lambda_k|\}^2$ satisfies $p_k \in (0, 1]$. That is, it holds that*

$$\left\| \boldsymbol{X}\boldsymbol{W} - \bar{\boldsymbol{X}} \right\|_F^2 \leq (1 - p_k) \left\| \boldsymbol{X} - \bar{\boldsymbol{X}} \right\|_F^2, \tag{6}$$

*for any $\boldsymbol{X} \in \mathbb{R}^{d \times k}$ where $\bar{\boldsymbol{X}} := \frac{1}{k}\boldsymbol{X}\mathbf{1}\mathbf{1}^\top$.*

Note that $\boldsymbol{W}$ is a $k \times k$ matrix, and $p_k$ depends on the number of active nodes $k$, but is independent of the total number of nodes $n$. Under Assumptions 1, 2, 3, 4, and 6, Theorem 1 provides the convergence rate of TELEPORTATION for any number of active nodes $k$.

**Theorem 1.** *Suppose that Assumptions 1, 2, 3, 4, and 6 hold. Let $\{\{\boldsymbol{x}_i^{(t)}\}_{i=1}^n\}_{t=0}^T$ denote the parameters generated by Alg. 1, and suppose that $\{\boldsymbol{x}_i^{(0)}\}_{i=1}^n$ is initialized with the same parameter $\bar{\boldsymbol{x}}^{(0)}$. Then, for any number of active nodes $k \in \{1, 2, \cdots, n\}$, there exists the step size $\eta$ such that $\frac{1}{T+1} \sum_{t=0}^T \mathbb{E}\|\nabla f(\bar{\boldsymbol{x}}_{active}^{(t)})\|^2$ is bounded from above by*

$$\mathcal{O}\left( \sqrt{\frac{Lr_0(\sigma^2 + (1 - \frac{k-1}{n-1})\zeta^2)}{kT}} + \left( \frac{L^2 r_0^2 (\sigma^2 + \zeta^2)(1 - p_k)}{T^2 p_k} \right)^{\frac{1}{3}} + \frac{Lr_0}{Tp_k} \right), \tag{7}$$

*where $\bar{\boldsymbol{x}}_{active}^{(t)} := \frac{1}{k} \sum_{v_i \in V_{active}^{(t)}} \boldsymbol{x}_i^{(t)}$ and $r_0 := f(\bar{\boldsymbol{x}}^{(0)}) - f^\star$.*

By tuning the number of active nodes $k$, we obtain the following theorem. To determine the proper number of active nodes $k$, the explicit form of $p_k$ is necessary. Here, we show the rates when we use a ring and exponential graph (Ying et al., 2021) as $\mathcal{G}_k$.

**Theorem 2.** *Suppose that Assumptions 1, 2, 3, 4, and 6 hold. Let $\{\{\boldsymbol{x}_i^{(t)}\}_{i=1}^n\}_{t=0}^T$ be the parameters generated by Alg. 1, and suppose that $\{\boldsymbol{x}_i^{(0)}\}_{i=1}^n$ is initialized with the same parameter $\bar{\boldsymbol{x}}^{(0)}$.*

***Ring:*** *Suppose that the active nodes are connected by a ring, i.e., $p_k = \Omega(k^{-2})$. Then, if we set the number of active nodes $k$ as follows:*

$$k = \max\left\{ 1, \min\left\{ \left\lceil \left( \frac{T(\sigma^2 + \zeta^2)}{Lr_0} \right)^{\frac{1}{7}} \right\rceil, n \right\} \right\},$$

*there exists $\eta$ such that $\frac{1}{T+1}\sum_{t=0}^{T}\mathbb{E}\|\nabla f(\bar{\boldsymbol{x}}_{active}^{(t)})\|^2$ is bounded from above by*

$$\mathcal{O}\left(\sqrt{\frac{Lr_0\sigma^2}{nT}} + \left(\frac{Lr_0(\sigma^2+\zeta^2)^{\frac{3}{4}}}{T}\right)^{\frac{4}{7}} + \left(\frac{Lr_0(\sigma^2+\zeta^2)^{\frac{2}{5}}}{T}\right)^{\frac{5}{7}} + \frac{Lr_0}{T}\right),$$

*where $\bar{\boldsymbol{x}}_{active}^{(t)} := \frac{1}{k}\sum_{v_i \in V_{active}^{(t)}}\boldsymbol{x}_i^{(t)}$ and $r_0 := f(\bar{\boldsymbol{x}}^{(0)}) - f^{\star}$.*

***Exp. Graph:*** *Suppose that the active nodes are connected by an exponential graph, i.e., $p_k = \Omega(\log_2^{-1}(k))$. Then, if we set the number of active nodes $k$ as follows:*

$$k = \max\left\{1, \min\left\{\left\lceil\left(\frac{T(\sigma^2+\zeta^2)}{Lr_0}\right)^{\frac{1}{3}}\right\rceil, \left\lceil\frac{T(\sigma^2+\zeta^2)}{Lr_0}\right\rceil, n\right\}\right\},$$

*there exists $\eta$ such that $\frac{1}{T+1}\sum_{t=0}^{T}\mathbb{E}\|\nabla f(\bar{\boldsymbol{x}}_{active}^{(t)})\|^2$ is bounded from above by*

$$\mathcal{O}\left(\sqrt{\frac{Lr_0\sigma^2}{nT}} + \left(\frac{L^2r_0^2(\sigma^2+\zeta^2)}{T^2}\log_2\left(\frac{T(\sigma^2+\zeta^2)}{Lr_0}\right)^{\frac{1}{3}}\right)^{\frac{1}{3}} + \frac{Lr_0}{T}\log_2\left(\frac{T(\sigma^2+\zeta^2)}{Lr_0}\right) + \frac{Lr_0}{T}\right).$$

All proofs are presented in Sec. B. The first term in Eq. (7) is worse than the first term in Eq. (5), but Theorem 2 indicates that the first term becomes the same as that of Decentralized SGD by carefully tuning $k$. For both cases, only the first term depends on the number of nodes $n$, which is improved as $n$ increases. Therefore, TELEPORTATION can completely eliminate the negative effect by increasing the number of nodes $n$, and its convergence rate is consistently improved as $n$ increases. As discussed in Sec. 2, several previous studies have attempted to avoid convergence rate degradation caused by large $n$ by designing topologies with large spectral gaps (Ying et al., 2021; Song et al., 2022; Takezawa et al., 2023b). However, no prior topology, except for a complete graph, can completely remove this degradation. By comparing with the complete graph, TELEPORTATION is more communication efficient. For instance, if a ring is used as $\mathcal{G}_k$, each node only needs to communicate with three nodes per iteration. Therefore, TELEPORTATION is the first method that does not suffer from large $n$ without sacrificing the communication efficiency.

### 3.3 EFFICIENT HYPERPARAMETER SEARCH FOR NUMBER OF ACTIVE NODES

TELEPORTATION can eliminate the convergence rate degradation caused by a large number of nodes $n$, while the number of active nodes $k \in \{1, \cdots, n\}$ needs to be tuned carefully. If $k$ is tuned by grid search, it requires $nT$ iterations in total. This hyperparameter-tuning becomes expensive as $n$ increases. To alleviate this issue, we further extend TELEPORTATION by proposing an efficient hyperparameter-tuning method for the number of active nodes $k$. The key idea for efficient hyperparameter-tuning is based on the following lemma.

**Lemma 1.** *Let $k^{\star} \in \{1, 2, 3, \cdots, n\}$ be the optimal number of active nodes. If $k^{\star} < n$, there exists $k \in \{1, 2, 4, 8, \cdots, 2^{\lfloor\log_2(n+1)\rfloor-1}\}$ that satisfies $\frac{k^{\star}}{4} < k \le k^{\star}$. Furthermore, it holds that $\sum_{i=0}^{\lfloor\log_2(n+1)\rfloor-1}2^i \le n$.*

---

**Algorithm 2** Efficient hyperparameter search for TELEPORTATION.

1: **Input:** the total number of iteration $2T$, and the number of nodes $n$.
2: run Alg. 1 for $T$ iterations with number of active nodes $n$, and let $\{\{\boldsymbol{x}_{n,i}^{(t)}\}_{i=1}^{n}\}_{t=0}^{T}$ denote the generated parameters.
3: **for** $k \in \{1, 2, 2^2, 2^3, \cdots, 2^{\lfloor\log_2(n+1)\rfloor-1}\}$ **in parallel do**
4:     run Alg. 1 for $T$ iterations with number of active nodes $k$.
5: **end for**
6: let $\{\{\boldsymbol{x}_{k,i}^{(t)}\}_{v_i \in V_{active}^{(t)}}\}_{t=0}^{T}$ denote the parameters generated by Alg. 1 with $k$ active nodes.
7: **return** the *best[b]* parameters among $k \in \{1, 2, 2^2, \cdots, 2^{\lfloor\log_2(n+1)\rfloor-1}, n\}$.

---

[b]Theoretically, we select the parameters with the smallest gradient norm as in Theorem 2. In practice, we can select $k$ that achieves the best accuracy on the validation datasets as in the vanilla grid search.

Lemma 1 shows that for any optimal number of active nodes $k^\star \leq n - 1$, a similar number of active nodes is contained in $\{1, 2, 2^2, \cdots, 2^{\lfloor \log_2(n+1) \rfloor - 1}\}$. This suggests that we only need to search from $\{1, 2, 2^2, \cdots, 2^{\lfloor \log_2(n+1) \rfloor - 1}, n\}$ to obtain the appropriate $k$ and we do not need run Alg. 1 for all $k \in \{1, 2, 3, \cdots, n\}$. Moreover, Lemma 1 indicates that Alg. 1 can be run with various $k \in \{1, 2, 2^2, \cdots, 2^{\lfloor \log_2(n+1) \rfloor - 1}\}$ in parallel because the total number of active nodes is less than or equal to $n$. Based on these observations, we propose an efficient hyperparameter-tuning method. The pseudo-code is shown in Alg. 2.

## 3.4 Convergence Analysis of Teleportation with Efficient Hyperparameter Search

Under the same assumptions as in Theorem 2, Theorem 3 provides the convergence rate of Alg. 2. The proof is deferred to Sec. C.

**Theorem 3.** *Suppose that Assumptions 1, 2, 3, 4, and 6 hold. Let $\{\{\boldsymbol{x}_{k,i}^{(t)}\}_{v_i \in V_{active}^{(t)}}\}_{t=0}^{T}$ denote the parameters of active nodes generated by Alg. 1 when the number of active nodes is set to $k$, and we define $\mathcal{K} := \{1, 2, 2^2, 2^3, \cdots, 2^{\lfloor \log_2(n+1) \rfloor - 1}, n\}$. Then, suppose that the parameters are initialized with the same parameter $\bar{\boldsymbol{x}}^{(0)}$.*

**Ring:** *If the active nodes are connected by a ring, i.e., $p_k = \Omega(k^{-2})$, there exists $\eta$ such that $\min_{k \in \mathcal{K}} \left( \frac{1}{T+1} \sum_{t=0}^{T} \mathbb{E} \| \nabla f(\bar{\boldsymbol{x}}_{active,k}^{(t)}) \|^2 \right)$ is bounded from above by*

$$\mathcal{O}\left( \sqrt{\frac{Lr_0\sigma^2}{nT}} + \left( \frac{Lr_0(\sigma^2 + \zeta^2)^{\frac{3}{4}}}{T} \right)^{\frac{4}{7}} + \left( \frac{Lr_0(\sigma^2 + \zeta^2)^{\frac{2}{5}}}{T} \right)^{\frac{5}{7}} + \frac{Lr_0}{T} \right),$$

*where $r_0 := f(\bar{\boldsymbol{x}}^{(0)}) - f^\star$ and $\bar{\boldsymbol{x}}_{active,k}^{(t)} := \frac{1}{k} \sum_{v_i \in V_{active}^{(t)}} \boldsymbol{x}_{k,i}^{(t)}$.*

**Exp. Graph:** *If the active nodes are connected by an exponential graph, i.e., $p_k = \Omega(\log_2^{-1} k)$, there exists $\eta$ such that $\min_{k \in \mathcal{K}} \left( \frac{1}{T+1} \sum_{t=0}^{T} \mathbb{E} \| \nabla f(\bar{\boldsymbol{x}}_{active,k}^{(t)}) \|^2 \right)$ is bounded from above by*

$$\mathcal{O}\left( \sqrt{\frac{Lr_0\sigma^2}{nT}} + \left( \frac{L^2 r_0^2(\sigma^2 + \zeta^2)}{T^2} \log_2 \left( \frac{T(\sigma^2 + \zeta^2)}{Lr_0} \right)^{\frac{1}{3}} \right)^{\frac{1}{3}} + \frac{Lr_0}{T} \log_2 \left( \frac{T(\sigma^2 + \zeta^2)}{Lr_0} \right) + \frac{Lr_0}{T} \right).$$

Theorem 3 shows that Alg. 2 can achieve exactly the same convergence rate as the one shown in Theorem 2 (i.e., the convergence rate with optimal $k$). Algorithm 2 requires only $2T$ iterations to find the proper number of active nodes, whereas grid search requires $nT$ iterations in total. Thus, Alg. 2 can find the appropriate number of active nodes more efficiently than the vanilla grid search.

## 4 Related Work

**Decentralized SGD and its Variants:** The literature on decentralized learning can be traced back to Tsitsiklis (1984), and Decentralized SGD (Lian et al., 2017) is currently the most widely used for reasons of its simplicity. Recently, many researchers have improved Decentralized SGD in various aspects. Pu & Nedić (2021), Yuan et al. (2019), Tang et al. (2018b), Yuan et al. (2021), Vogels et al. (2021), Takezawa et al. (2023a), Aketi et al. (2023), and Di et al. (2024) proposed decentralized learning methods that are robust to data heterogeneity. Tang et al. (2018a), Koloskova et al. (2019), Vogels et al. (2020), Kovalev et al. (2021), and Zhao et al. (2022) proposed communication compression methods. Liu et al. (2022) analyzed Decentralized SGD with client sampling.

**Token Algorithm:** Token algorithms (Johansson et al., 2007; Dorfman & Levy, 2022; Even, 2023; Hendrikx, 2023) are a variant of decentralized learning methods different from consensus-based decentralized learning methods, such as Decentralized SGD. In contrast to the consensus-based decentralized learning methods, a parameter called token randomly walks on the topology and is updated on each node. Since token algorithms have only one parameter, they do not suffer from the issue of parameters held by each node drifting away. However, their convergence rates cannot

achieve linear speedup as that in Decentralized SGD (Even, 2023). In TELEPORTATION, $k$ parameters randomly move from previous active nodes to the next active nodes and are updated on the active nodes. Then, by carefully selecting $k$, TELEPORTATION can alleviate the issue of parameter drifting without sacrificing linear speedup property.

**Client Sampling:** Client sampling is widely studied in federated learning to reduce the communication costs between the central server and nodes (McMahan et al., 2017; Fraboni et al., 2021; Wu et al., 2023). McMahan et al. (2017) proposed sampling a subset of nodes randomly. Cho et al. (2020) proposed selecting nodes according to the loss values. Chen et al. (2022) and Wang et al. (2023) proposed sampling nodes according to their gradient norms. In decentralized learning, Liu et al. (2022) studied client sampling and analyzed the convergence rate. However, as discussed in Sec. 3.1, the client sampling cannot alleviate the degradation of the convergence rate caused by a large number of nodes.

## 5 EXPERIMENT

### 5.1 SYNTHETIC EXPERIMENT

**Setting:** We followed the experimental setting in Koloskova et al. (2020b) and set the number of nodes $n$ to 100 and loss function as $f_i(\boldsymbol{x}) \coloneqq \frac{1}{2}\|\boldsymbol{A}_i(\boldsymbol{x} - \boldsymbol{b}_i)\|^2$ with $\boldsymbol{A}_i \coloneqq \frac{i}{\sqrt{n}}\boldsymbol{I}_d$ and $d = 50$. $\boldsymbol{b}_i$ was drawn from $\mathcal{N}(\boldsymbol{0}, \frac{\zeta^2}{i^2}\boldsymbol{I}_d)$ for each node. We defined the stochastic gradient as $\nabla F_i(\boldsymbol{x}; \xi) \coloneqq \nabla f_i(\boldsymbol{x}) + \epsilon$ where $\epsilon$ was drawn from $\mathcal{N}(\boldsymbol{0}, \frac{\sigma^2}{d}\boldsymbol{I}_d)$ at each iteration. We used a ring and Base-2 Graph (Takezawa et al., 2023b) as the topology. The ring is one of the most commonly used topologies for decentralized learning. The Base-2 Graph is the state-of-the-art topology for Decentralized SGD, and Takezawa et al. (2023b) demonstrated that the Base-2 Graph can be superior to the various topologies, e.g., an exponential graph, 1-peer exponential graph (Ying et al., 2021), and EquiTopo (Song et al., 2022). Note that the Base-2 Graph assumes that any two nodes can communicate as in TELEPORTATION. For each setting, we tuned the step size to reach the target accuracy as few iterations as possible. See Sec. G for a more detailed setting.

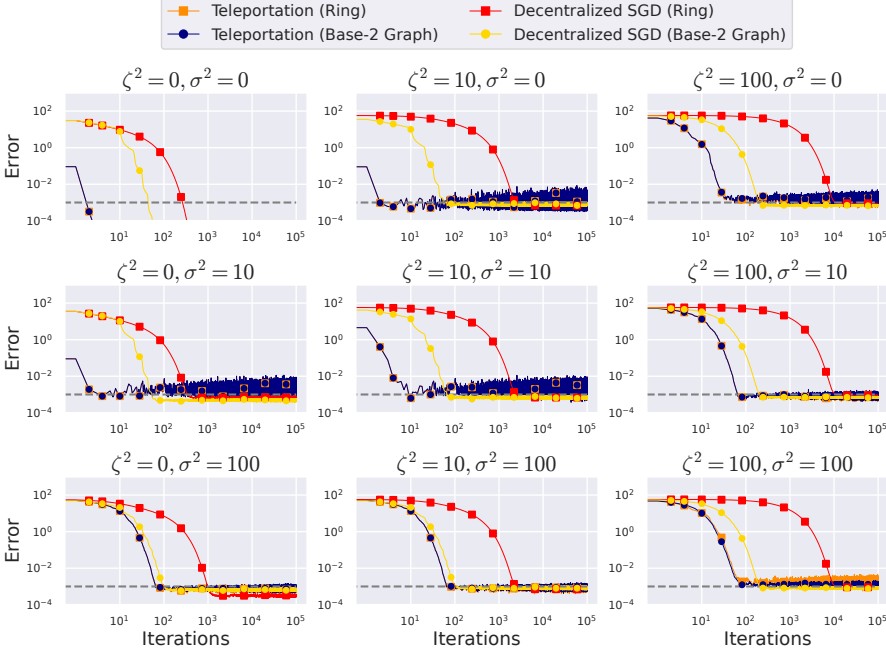

Figure 2: Convergence of the error to the target accuracy 0.001 for different stochastic noise $\sigma^2$ and heterogeneity $\zeta^2$. We plotted $\frac{1}{k}\sum_{v_i \in V_{\text{active}}^{(t)}} \|\boldsymbol{x}_i^{(t)} - \boldsymbol{x}^\star\|^2$ and $\frac{1}{n}\sum_{i=1}^{n} \|\boldsymbol{x}_i^{(t)} - \boldsymbol{x}^\star\|^2$ as the error for TELEPORTATION and Decentralized SGD, respectively. TELEPORTATION consistently reached the target accuracy faster than Decentralized SGD.

**Results:** We depict the results in Fig. 2. For all cases, TELEPORTATION required fewer iterations to reach the target accuracy than Decentralized SGD. By comparing Decentralized SGD with a ring, TELEPORTATION converged up to three orders of magnitude faster than Decentralized SGD. Even in the case when the state-of-the-art topology, Base-2 Graph, is used, TELEPORTATION converged up to 10 times faster than

Table 1: The number of active nodes $k$ selected by Alg. 2 in Fig. 2. The left value is the number of active nodes when the topology is a ring, and the right value is the number when the topology is the Base-2 Graph.

|  | $\zeta^2 = 0$ | $\zeta^2 = 10$ | $\zeta^2 = 100$ |
|---|---|---|---|
| $\sigma^2 = 0$ | 1 / 1 | 1 / 1 | 4 / 4 |
| $\sigma^2 = 10$ | 1 / 1 | 1 / 1 | 8 / 8 |
| $\sigma^2 = 100$ | 8 / 8 | 8 / 8 | 4 / 16 |

Decentralized SGD. Table 1 lists the number of active nodes $k$ selected by Alg. 2. Although the total number of nodes $n$ is 100, a small number of nodes $k$ was selected. Therefore, activating only a few nodes can prevent the parameters from being far away and lead to a faster convergence rate than that of Decentralized SGD.

## 5.2 NEURAL NETWORKS

**Setting:** We used Fashion MNIST (Xiao et al., 2017) and CIFAR-10 (Krizhevsky, 2009) as datasets and used LeNet (LeCun et al., 1998) and VGG (Simonyan & Zisserman, 2015) as neural networks, respectively. To use the momentum in TELEPORTATION, the momentum is copied from the previous active nodes to the next active nodes, as well as the parameters. We set the number of nodes $n$ to 25 and distributed the data to nodes using Dirichlet distribution with parameter $\alpha$ (Hsu et al., 2019). As $\alpha$ approaches zero, each node comes to have a different dataset. We repeated all experiments with three different seed values, reporting the averages. See Sec. G for a more detailed setting.

**Results:** We depict the results in Fig. 3. When $\alpha = 0.1$, TELEPORTATION outperformed Decentralized SGD and trained the neural networks more stably. When $\alpha = 10.0$, all comparison methods achieved a competitive accuracy. By comparing the results with $\alpha = 0.1$ and $10.0$, the accuracy curves of Decentralized SGD became unstable in the heterogeneous setting, whereas the accuracy curves of TELEPORTATION were stable in both cases. This is because Decentralized SGD suffers from client drift, while TELEPORTATION can suppress it by initializing the parameters of active nodes by using the parameters of other nodes.

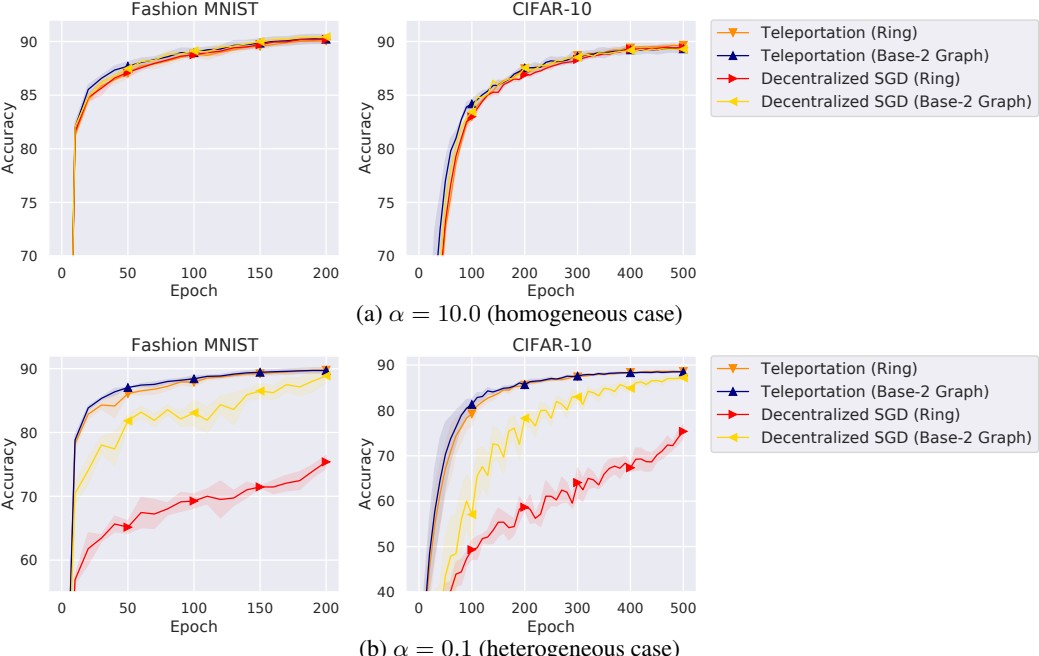

Figure 3: Test accuracy of TELEPORTATION and Decentralized SGD for different heterogeneity. All methods achieved competitive accuracy in the homogeneous case, while TELEPORTATION outperformed Decentralized SGD in the heterogeneous case.

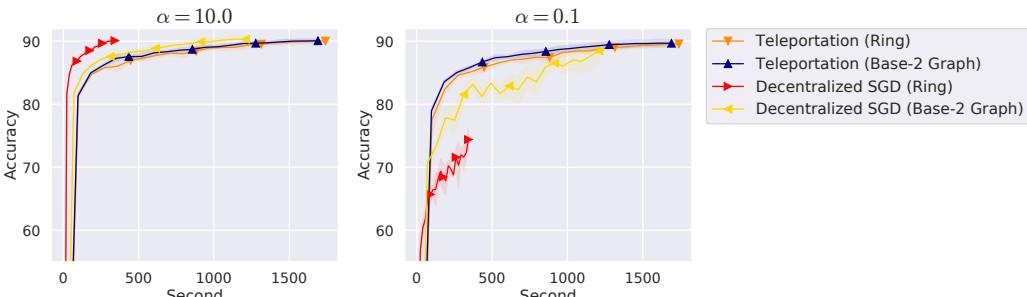

Figure 4: Test accuracy of TELEPORTATION and Decentralized SGD under the heterogeneous networks with $\tau = 5$. Decentralized SGD with the ring reached a high accuracy faster than the other methods in the homogeneous case, while TELEPORTATION reached a high accuracy faster in the heterogeneous case. Note that the number of epochs was set the same for all methods.

## 5.3 COMPARISON UNDER HETEROGENEOUS NETWORKS

Next, we examine the effectiveness of TELEPORTATION in terms of wallclock time. In TELEPORTATION, active nodes send the parameters to the next active nodes, which are randomly sampled from the entire set of nodes. Thus, the communication may happen between distant nodes, e.g., nodes located in distant regions. In this section, we evaluate TELEPORTATION under the heterogeneous networks where communication costs differ between pairs of nodes.

**Setting:** We used Fashion MNIST and LeNet as with Sec. 5.2. To simulate a heterogeneous network, we added $\tau \times \text{mod}(|i - j|, 24)\,\mu\text{s}$ delay when nodes $i$ and $j$ communicate, where $\text{mod}(a, b)$ is the remainder of dividing $a$ by $b$. We constructed a ring by connecting node $i$ to node $(1 + \text{mod}(i, 25))$. This ring is the optimal topology in terms of the communication delay, and the communication in Decentralized SGD with the ring was delayed by $\tau\,\mu\text{s}$, whereas the communication in TELEPORTATION was delayed by at most $24\tau\,\mu\text{s}$. The communication in Decentralized SGD with the Base-2 Graph was also delayed by at most $24\tau\,\mu\text{s}$ since the Base-2 Graph assumes that any two nodes can communicate, and communication occurs between various nodes.

**Results:** We depict the results with $\tau = 5$ in Fig. 4. In the Appendix, we also show the results with $\tau = 0$ in Fig. 7. Decentralized SGD with the ring finished the training faster than Decentralized SGD with the Base-2 Graph and TELEPORTATION. However, when $\alpha = 0.1$, the accuracy of Decentralized SGD with the ring increased very slowly, and TELEPORTATION reached a high accuracy faster than Decentralized SGD. When $\alpha = 10.0$, Decentralized SGD with the ring reached a high accuracy faster than the other methods. Therefore, Decentralized SGD with a sparse topology is the preferred method when the data distributions are homogeneous, while when the data distributions are heterogeneous, TELEPORTATION can be a preferred method even if the network is heterogeneous since the methods that can prevent the parameter drifting are necessary to achieve high accuracy.

## 6 CONCLUSION AND LIMITATION

In this paper, we propose TELEPORTATION, which activates a subset of nodes and performs gossip averaging on a relatively small topology comprising only active nodes. We showed that TELEPORTATION converges to the stationary point without suffering from a large number of nodes by activating a proper number of nodes. Furthermore, we proposed an efficient hyperparameter tuning method to search for this appropriate number of nodes. We experimentally investigated the effectiveness of TELEPORTATION, demonstrating that TELEPORTATION can converge faster than Decentralized SGD and train neural networks more stably when the data distributions are heterogeneous.

**Limitation:** TELEPORTATION is not applicable in the case that there exists a pair of nodes that cannot communicate. Extending TELEPORTATION to such a setting is one of the most promising future directions. Furthermore, since active nodes are randomly sampled from the entire set of nodes, communication may happen between distant nodes in the heterogeneous networks. In Sec. 5.3, we demonstrated that when the data distributions are heterogeneous, TELEPORTATION can be a preferred method even if the network is heterogeneous, while it would be a promising future direction to ease this condition to prevent nodes from communicating with distant nodes.

ACKNOWLEDGMENTS

This work was supported by JSPS KAKENHI Grant Number 23KJ1336. We thank Anastasia Koloskova for her helpful comments on the practical implementation of TELEPORTATION described in Sec. A.

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

## A  TELEPORTATION WITH COMMUNICATION OVERLAP

In the implementation shown in Alg. 1, line 7 in Alg. 1 does not start until line 12 is completed. We can modify Alg. 1 so that the exchanges of parameters in lines 7 and 12 are performed simultaneously. We show the pseudo-code and illustration in Alg. 3 and Fig. 5. The communication in lines 7 and 12 in Alg. 1 corresponds to that in lines 9 and 13 in Alg. 3.

---

**Algorithm 3** TELEPORTATION

---

1: **Input:** the number of nodes $n$, set of nodes $V_n$, number of active nodes $k$, total number of iteration $T$, step size $\eta$, and topology comprising $k$ active nodes $\mathcal{G}_k = (\{1, \cdots, k\}, E)$.

2: sample active $k$ nodes $V_{\text{active}}^{(0)}$ from $V_n$ without replacement.

3: assign $\{1, 2, \cdots, k\}$ to variables $\{\texttt{token\_id}_i^{(0)} \mid v_i \in V_{\text{active}}^{(0)}\}$ randomly without overlap.

4: **for** $t \in \{0, 1, \cdots, T\}$ **do**

5:     sample next active $k$ nodes $V_{\text{active}}^{(t+1)}$ from $V_n$ without replacement, and assign $\{1, 2, \cdots, k\}$ to variables $\{\texttt{token\_id}_i^{(t+1)} \mid v_i \in V_{\text{active}}^{(t+1)}\}$ randomly without overlap.

6:     **for** $i \in \{1, 2, \cdots, n\}$ **in parallel do**

7:         **if** $v_i \in V_{\text{active}}^{(t)}$ **then**[c]

8:             $\boldsymbol{y}_i^{(t)} = \boldsymbol{x}_i^{(t)} - \eta \nabla F_i(\boldsymbol{x}_i^{(t)}; \xi_i^{(t)})$.

9:             send $\boldsymbol{y}_i^{(t)}$ to $v_j \in \{v_j \in V_{\text{active}}^{(t+1)} \mid (\texttt{token\_id}_i^{(t)}, \texttt{token\_id}_j^{(t+1)}) \in E\}$.

10:        **end if**

11:        **if** $v_i \in V_{\text{active}}^{(t+1)}$ **then**

12:            receive $\boldsymbol{y}_j^{(t)}$ from $v_j \in \{v_j \in V_{\text{active}}^{(t)} \mid (\texttt{token\_id}_i^{(t+1)}, \texttt{token\_id}_j^{(t)}) \in E\}$.

13:            $\boldsymbol{x}_i^{(t+1)} = \sum_{v_j \in V_{\text{active}}^{(t)}} W_{\texttt{token\_id}_i^{(t+1)}, \texttt{token\_id}_j^{(t)}} \boldsymbol{y}_j^{(t)}$.

14:        **end if**

15:    **end for**

16: **end for**

---

[c]The update rule of the parameters of the inactive nodes is not described because the parameters held by the inactive nodes are discarded and initialized with the parameters of the other active nodes when they are activated in line 13. The parameters of inactive nodes do not affect the behavior of active node parameters.

---

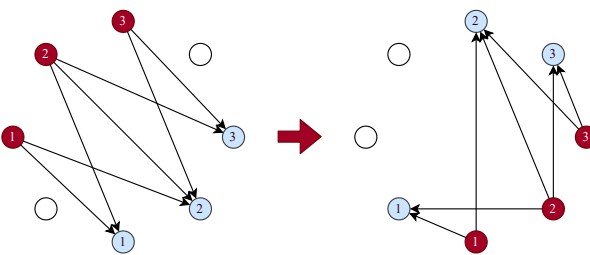

Figure 5: Illustration of Alg. 3 with $n = 8$ and $k = 3$. We use a line as the topology consisting of active nodes $\mathcal{G}_k = (\{1, 2, 3\}, \{(1, 1), (1, 2), (2, 2), (2, 3), (3, 3)\})$. The black nodes represent active nodes, and the number written on the node is $\texttt{token\_id}_i^{(t)}$. The blue nodes represent the next active nodes, and the number on the node is $\texttt{token\_id}_i^{(t+1)}$.

## B  PROOF OF THEOREMS 1 AND 2

### B.1  USEFUL INEQUALITY

**Lemma 2.** *For any $a, b \in \mathbb{R}^d$, it holds that*

$$\|a + b\|^2 \le (1 + \gamma)\|a\|^2 + (1 + \frac{1}{\gamma})\|b\|^2, \tag{8}$$

*for any $\gamma > 0$.*

**Lemma 3.** *For any $a, b \in \mathbb{R}^d$, it holds that*

$$2\langle a, b \rangle \le \|a\|^2 + \|b\|^2. \tag{9}$$

### B.2  NOTATION

In this section, we introduce the notation used in the proof. TELEPORTATION has $n$ parameters, $\{x_i^{(t)}\}_{i=1}^n$, while the parameters of inactive nodes are not used and do not affect the behavior of the parameters of active nodes. To simplify the notation, we introduce the variables $\{z_m^{(t)}\}_{m=1}^k$ that correspond to the active nodes parameters.

From line 5 in Alg. 3, $\texttt{text\_id}_i^{(t)}$ has a different value for each active node $v_i \in V_{\text{active}}^{(t)}$. That is, there is one-to-one correspondence between $\{i \mid v_i \in V_{\text{active}}^{(t)}\}$ and $\{1, 2, \cdots, k\}$. Let $g^{(t)} : \{i \mid v_i \in V_{\text{active}}^{(t)}\} \to \{1, 2, \cdots, k\}$ be the function that takes node index $i$ and returns $\texttt{token\_id}_i^{(t)}$. Since $g^{(t)}$ is a bijection function, there exists an inverse function $(g^{(t)})^{-1} : \{1, 2, \cdots, k\} \to \{i \mid v_i \in V_{\text{active}}^{(t)}\}$. Using this inverse function, we define variable $\texttt{node\_id}_m^{(t)}$ as follows:

$$\texttt{node\_id}_m^{(t)} := (g^{(t)})^{-1}(m),$$

for any $m \in \{1, 2, \cdots, k\}$. $\texttt{node\_id}_m^{(t)}$ stores the node index $i$ whose $\texttt{token\_id}_i^{(t)}$ stores $m$. Using this notation, we define $z_m^{(t)} \in \mathbb{R}^d$ as follows:

$$z_m^{(t)} = x_{\texttt{node\_id}_m^{(t)}}^{(t)},$$

for all $m \in \{1, 2, \cdots, k\}$ and $t$. From the definition of $z_m^{(t)}$, we can get its update rule as follows:

$$
\begin{aligned}
z_m^{(t+1)} &= x_{\texttt{node\_id}_m^{(t+1)}}^{(t)} \\
&= \sum_{v_j \in V_{\text{active}}^{(t)}} W_{\texttt{token\_id}_{\texttt{node\_id}_m^{(t+1)}}^{(t+1)}, \texttt{token\_id}_j^{(t)}} \left( x_j^{(t)} - \eta \nabla F_j(x_j^{(t)}; \xi_j^{(t)}) \right) \\
&= \sum_{v_j \in V_{\text{active}}^{(t)}} W_{m, \texttt{token\_id}_j^{(t)}} \left( x_j^{(t)} - \eta \nabla F_j(x_j^{(t)}; \xi_j^{(t)}) \right) \\
&= \sum_{l=1}^k W_{m,l} \left( x_{\texttt{node\_id}_l^{(t)}}^{(t)} - \eta \nabla F_{\texttt{node\_id}_l^{(t)}}(x_{\texttt{node\_id}_l^{(t)}}^{(t)}; \xi_{\texttt{node\_id}_l^{(t)}}^{(t)}) \right) \\
&= \sum_{l=1}^k W_{m,l} \left( z_l^{(t)} - \eta \nabla F_{\texttt{node\_id}_l^{(t)}}(z_l^{(t)}; \xi_{\texttt{node\_id}_l^{(t)}}^{(t)}) \right).
\end{aligned}
$$

In the next section, we analyze the convergence behavior of $\{z_m^{(t)}\}_{m=1}^k$. Note that sets $\{z_m^{(t)}\}_{m=1}^k$ and $\{x_i^{(t)} \mid v_i \in V_{\text{active}}^{(t)}\}$ are equivalent, and their averages are equivalent, i.e.,

$$\frac{1}{k} \sum_{m=1}^k z_m^{(t)} = \frac{1}{k} \sum_{m=1}^k x_{\texttt{node\_id}_m^{(t)}}^{(t)} = \frac{1}{k} \sum_{i=1}^n x_i^{(t)} \mathbf{1}_{v_i \in V_{\text{active}}^{(t)}} = \frac{1}{k} \sum_{v_i \in V_{\text{active}}^{(t)}} x_i^{(t)}, \tag{10}$$

where $\mathbf{1}_{v_i \in V_{\text{active}}^{(t)}}$ is an indicator function. Furthermore, let $\boldsymbol{Z} \in \mathbb{R}^{d \times k}$, $\boldsymbol{G} \in \mathbb{R}^{d \times k}$, and $\nabla f(\boldsymbol{Z}) \in \mathbb{R}^{d \times k}$ as follows:

$$\boldsymbol{Z}^{(t)} := \left( \boldsymbol{z}_1^{(t)}, \boldsymbol{z}_2^{(t)}, \cdots, \boldsymbol{z}_k^{(t)} \right),$$

$$\boldsymbol{G}^{(t)} := \left( \nabla F_{\text{node\_id}_1^{(t)}}(\boldsymbol{z}_1^{(t)}; \xi_{\text{node\_id}_1^{(t)}}^{(t)}), \cdots, \nabla F_{\text{node\_id}_k^{(t)}}(\boldsymbol{z}_k^{(t)}; \xi_{\text{node\_id}_k^{(t)}}^{(t)}) \right),$$

$$\nabla f(\boldsymbol{Z}^{(t)}) := \left( \nabla f(\boldsymbol{z}_1^{(t)}), \nabla f(\boldsymbol{z}_2^{(t)}), \cdots, \nabla f(\boldsymbol{z}_k^{(t)}) \right).$$

By using the above notation, Alg. 3 can be rewritten as follows:

$$\boldsymbol{Z}^{(t+1)} = \left( \boldsymbol{Z}^{(t)} - \eta \boldsymbol{G}^{(t)} \right) \boldsymbol{W}^\top.$$

## B.3 MAIN PROOF

**Lemma 4.** *Suppose that Assumptions 2, 3, 4, and 6 hold. If the step size satisfies $\eta \leq \frac{1}{4L}$, then it holds that*

$$\mathbb{E}f(\bar{\boldsymbol{z}}^{(t+1)}) \leq \mathbb{E}f(\bar{\boldsymbol{z}}^{(t)}) - \frac{\eta}{4}\mathbb{E}\left\| \nabla f(\bar{\boldsymbol{z}}^{(t)}) \right\|^2 + L^2 \eta \Xi^{(t)} + \frac{L\sigma^2}{2k}\eta^2 + \frac{L\zeta^2}{2k}\left( 1 - \frac{k-1}{n-1} \right)\eta^2,$$

*where $\bar{\boldsymbol{z}}^{(t)} := \frac{1}{k}\sum_{i=1}^k \boldsymbol{z}_i^{(t)}$ and $\Xi^{(t)} := \frac{1}{k}\sum_{i=1}^k \mathbb{E}\|\boldsymbol{z}_i^{(t)} - \bar{\boldsymbol{z}}^{(t)}\|^2$.*

*Proof.* By calculating the average of the update rule of $\boldsymbol{z}_i^{(t)}$, we obtain

$$\bar{\boldsymbol{z}}^{(t+1)} = \frac{1}{k}\sum_{m=1}^k \sum_{l=1}^k W_{m,l}\left( \boldsymbol{z}_l^{(t)} - \eta \nabla F_{\text{node\_id}_l^{(t)}}(\boldsymbol{z}_l^{(t)}; \xi_{\text{node\_id}_l^{(t)}}^{(t)}) \right)$$

$$= \bar{\boldsymbol{z}}^{(t)} - \frac{\eta}{k}\sum_{l=1}^k \nabla F_{\text{node\_id}_l^{(t)}}(\boldsymbol{z}_l^{(t)}; \xi_{\text{node\_id}_l^{(t)}}^{(t)}),$$

where we use $\sum_{m=1}^k W_{m,l} = 1$. Using Assumption 2, we get

$$\mathbb{E}_{t+1}f(\bar{\boldsymbol{z}}^{(t+1)})$$

$$= \mathbb{E}_{t+1}f\left( \bar{\boldsymbol{z}}^{(t)} - \frac{\eta}{k}\sum_{j=1}^k \nabla F_{\text{node\_id}_j^{(t)}}(\boldsymbol{z}_j^{(t)}; \xi_{\text{node\_id}_j^{(t)}}^{(t)}) \right)$$

$$\leq f(\bar{\boldsymbol{z}}^{(t)}) - \eta \left\langle \nabla f(\bar{\boldsymbol{z}}^{(t)}), \frac{1}{k}\sum_{j=1}^k \mathbb{E}_{t+1}\nabla F_{\text{node\_id}_j^{(t)}}(\boldsymbol{z}_j^{(t)}; \xi_{\text{node\_id}_j^{(t)}}^{(t)}) \right\rangle$$

$$+ \frac{L\eta^2}{2}\mathbb{E}_{t+1}\left\| \frac{1}{k}\sum_{j=1}^k \nabla F_{\text{node\_id}_j^{(t)}}(\boldsymbol{z}_j^{(t)}; \xi_{\text{node\_id}_j^{(t)}}^{(t)}) \right\|^2$$

$$= f(\bar{\boldsymbol{z}}^{(t)}) \underbrace{- \eta \left\langle \nabla f(\bar{\boldsymbol{z}}^{(t)}), \frac{1}{k}\sum_{j=1}^k \nabla f(\boldsymbol{z}_j^{(t)}) \right\rangle}_{T_1} + \underbrace{\frac{L\eta^2}{2}\mathbb{E}_{t+1}\left\| \frac{1}{k}\sum_{j=1}^k \nabla F_{\text{node\_id}_j^{(t)}}^{(t)}(\boldsymbol{z}_j^{(t)}; \xi_{\text{node\_id}_j^{(t)}}^{(t)}) \right\|^2}_{T_2},$$

where we use the fact that active nodes are randomly selected, i.e., $\texttt{node\_id}_j^{(t)}$ is randomly assigned to $\{1, 2, \cdots, n\}$, in the last equality. $T_1$ and $T_2$ are bounded as follows:

$$-T_1 = -\left\| \nabla f(\bar{z}^{(t)}) \right\|^2 + \left\langle \nabla f(\bar{z}^{(t)}), \frac{1}{k} \sum_{j=1}^{k} \nabla f(z_j^{(t)}) - \nabla f(\bar{z}^{(t)}) \right\rangle$$

$$\leq -\frac{1}{2} \left\| \nabla f(\bar{z}^{(t)}) \right\|^2 + \frac{1}{2} \left\| \frac{1}{k} \sum_{j=1}^{k} \nabla f(z_j^{(t)}) - \nabla f(\bar{z}^{(t)}) \right\|^2$$

$$\leq -\frac{1}{2} \left\| \nabla f(\bar{z}^{(t)}) \right\|^2 + \frac{1}{2k} \sum_{j=1}^{k} \left\| \nabla f(z_j^{(t)}) - \nabla f(\bar{z}^{(t)}) \right\|^2$$

$$\leq -\frac{1}{2} \left\| \nabla f(\bar{z}^{(t)}) \right\|^2 + \frac{L^2}{2k} \sum_{j=1}^{k} \left\| z_j^{(t)} - \bar{z}^{(t)} \right\|^2.$$

$$T_2 = \mathbb{E}_{t+1} \left\| \frac{1}{k} \sum_{j=1}^{k} \nabla f_{\texttt{node\_id}_j^{(t)}}(z_j^{(t)}) \right\|^2 + \mathbb{E}_{t+1} \left\| \frac{1}{k} \sum_{j=1}^{k} \nabla F_{\texttt{node\_id}_j^{(t)}}(z_j^{(t)}; \xi_{\texttt{node\_id}_j^{(t)}}^{(t)}) - \nabla f_{\texttt{node\_id}_j^{(t)}}(z_j^{(t)}) \right\|^2$$

$$\leq \mathbb{E}_{t+1} \left\| \frac{1}{k} \sum_{j=1}^{k} \nabla f_{\texttt{node\_id}_j^{(t)}}(z_j^{(t)}) \right\|^2 + \frac{\sigma^2}{k}$$

$$\leq \left\| \frac{1}{k} \sum_{j=1}^{k} \nabla f(z_j^{(t)}) \right\|^2 + \mathbb{E}_{t+1} \left\| \frac{1}{k} \sum_{j=1}^{k} \nabla f_{\texttt{node\_id}_j^{(t)}}(z_j^{(t)}) - \nabla f(z_j^{(t)}) \right\|^2 + \frac{\sigma^2}{k}$$

$$\leq \left\| \frac{1}{k} \sum_{j=1}^{k} \nabla f(z_j^{(t)}) \right\|^2 + \frac{\zeta^2}{k} \left( 1 - \frac{k-1}{n-1} \right) + \frac{\sigma^2}{k}$$

$$\leq 2 \left\| \frac{1}{k} \sum_{j=1}^{k} \nabla f(z_j^{(t)}) - \nabla f(\bar{z}^{(t)}) \right\|^2 + 2 \left\| \nabla f(\bar{z}^{(t)}) \right\|^2 + \frac{\zeta^2}{k} \left( 1 - \frac{k-1}{n-1} \right) + \frac{\sigma^2}{k}$$

$$\leq \frac{2L^2}{k} \sum_{j=1}^{k} \left\| z_j^{(t)} - \bar{z}^{(t)} \right\|^2 + 2 \left\| \nabla f(\bar{z}^{(t)}) \right\|^2 + \frac{\zeta^2}{k} \left( 1 - \frac{k-1}{n-1} \right) + \frac{\sigma^2}{k},$$

where we use the fact that the active nodes $V_{\text{active}}^{(t)}$ are sampled from $V_n$ without replacement in the third inequality. Using the above inequalities, we obtain

$$\mathbb{E}_{t+1} f(\bar{z}) \leq f(\bar{z}^{(t)}) - \frac{\eta}{2} \left\| \nabla f(\bar{z}^{(t)}) \right\|^2 + \frac{L^2 \eta}{2k} \sum_{j=1}^{k} \left\| z_j^{(t)} - \bar{z}^{(t)} \right\|^2$$

$$+ \frac{L^3 \eta^2}{k} \sum_{j=1}^{k} \left\| z_j^{(t)} - \bar{z}^{(t)} \right\|^2 + L\eta^2 \|\nabla f(\bar{z}^{(t)})\|^2 + \frac{L\zeta^2}{2k} \left( 1 - \frac{k-1}{n-1} \right) \eta^2 + \frac{L\sigma^2}{2k} \eta^2.$$

Using $\eta \leq \frac{1}{4L}$, we obtain the desired result. $\qquad \square$

**Lemma 5.** *Suppose that Assumptions 2, 3, 4, and 6 hold. If the step size satisfies $\eta \leq \frac{p_k}{\sqrt{24}L}$, then it holds that*

$$\Xi^{(t+1)} \leq (1 - \frac{p_k}{4})\Xi^{(t)} + \frac{6(1-p_k)\eta^2}{p_k} \mathbb{E} \left\| \nabla f(\bar{z}^{(t)}) \right\|^2 + (1-p_k)(\sigma^2 + \zeta^2)\eta^2,$$

*where $\bar{z}^{(t)} := \frac{1}{k} \sum_{i=1}^{k} z_i^{(t)}$ and $\Xi^{(t)} := \frac{1}{k} \sum_{i=1}^{k} \mathbb{E}\|z_i^{(t)} - \bar{z}^{(t)}\|^2$.*

*Proof.*

$$\mathbb{E}_{t+1} \left\| \boldsymbol{Z}^{(t+1)} - \bar{\boldsymbol{Z}}^{(t+1)} \right\|_F^2$$

$$= \mathbb{E}_{t+1} \left\| (\boldsymbol{Z}^{(t)} - \eta \boldsymbol{G}^{(t)}) \boldsymbol{W}^\top - (\bar{\boldsymbol{Z}}^{(t)} - \eta \bar{\boldsymbol{G}}^{(t)}) \right\|_F^2$$

$$\leq (1 - p_k) \mathbb{E}_{t+1} \left\| (\boldsymbol{Z}^{(t)} - \eta \boldsymbol{G}^{(t)}) - (\bar{\boldsymbol{Z}}^{(t)} - \eta \bar{\boldsymbol{G}}^{(t)}) \right\|_F^2$$

$$\leq (1 - p_k) \mathbb{E}_{t+1} \left\| (\boldsymbol{Z}^{(t)} - \eta \boldsymbol{G}^{(t)}) - \bar{\boldsymbol{Z}}^{(t)} \right\|_F^2$$

$$\leq (1 - p_k) \left\| (\boldsymbol{Z}^{(t)} - \eta \nabla f(\boldsymbol{Z}^{(t)})) - \bar{\boldsymbol{Z}}^{(t)} \right\|_F^2 + (1 - p_k) \eta^2 \mathbb{E}_{t+1} \left\| \boldsymbol{G}^{(t)} - f(\boldsymbol{Z}^{(t)}) \right\|^2$$

$$\leq (1 - p_k) \left\| (\boldsymbol{Z}^{(t)} - \eta \nabla f(\boldsymbol{Z}^{(t)})) - \bar{\boldsymbol{Z}}^{(t)} \right\|_F^2 + (1 - p_k) k (\sigma^2 + \zeta^2) \eta^2$$

$$\leq (1 - \frac{p_k}{2}) \left\| \boldsymbol{Z}^{(t)} - \bar{\boldsymbol{Z}}^{(t)} \right\|_F^2 + \frac{3(1 - p_k) \eta^2}{p_k} \underbrace{\left\| \nabla f(\boldsymbol{Z}^{(t)}) \right\|_F^2}_{T} + (1 - p_k) k (\sigma^2 + \zeta^2) \eta^2.$$

$T$ is bounded as follows:

$$T \leq 2 \left\| \nabla f(\boldsymbol{Z}^{(t)}) - \nabla f(\bar{\boldsymbol{Z}}^{(t)}) \right\|_F^2 + 2 \left\| \nabla f(\bar{\boldsymbol{Z}}^{(t)}) \right\|_F^2$$

$$\leq 2L^2 \left\| \boldsymbol{Z}^{(t)} - \bar{\boldsymbol{Z}}^{(t)} \right\|_F^2 + 2k \left\| \nabla f(\bar{\boldsymbol{z}}^{(t)}) \right\|^2,$$

where $\bar{\boldsymbol{Z}}^{(t)} := \frac{1}{k} \boldsymbol{Z}^{(t)} \mathbf{1} \mathbf{1}^\top$. Then, we get

$$\mathbb{E}_{t+1} \left\| \boldsymbol{Z}^{(t+1)} - \bar{\boldsymbol{Z}}^{(t+1)} \right\|_F^2$$

$$\leq (1 - \frac{p_k}{2}) \left\| \boldsymbol{Z}^{(t)} - \bar{\boldsymbol{Z}}^{(t)} \right\|_F^2 + \frac{6(1 - p_k) L^2 \eta^2}{p_k} \left\| \boldsymbol{Z}^{(t)} - \bar{\boldsymbol{Z}}^{(t)} \right\|_F^2$$

$$+ \frac{6(1 - p_k) k \eta^2}{p_k} \left\| \nabla f(\bar{\boldsymbol{z}}^{(t)}) \right\|^2 + (1 - p_k) k (\sigma^2 + \zeta^2) \eta^2.$$

Using $\eta \leq \frac{p_k}{\sqrt{24} L}$, we obtain the desired result. $\qquad \square$

**Lemma 6.** *Suppose that Assumptions 2, 3, 4, and 6 hold, and $\{\boldsymbol{x}_i^{(0)}\}_{i=1}^n$ is initialized with the same parameter $\bar{\boldsymbol{x}}^{(0)}$. If the step size satisfies $\eta \leq \frac{p_k}{\sqrt{24} L}$, then it holds that*

$$\frac{1}{T+1} \sum_{t=0}^{T} \Xi^{(t)} \leq \frac{24(1 - p_k)}{p_k^2 (T+1)} \eta^2 \sum_{t=0}^{T} \mathbb{E} \| \nabla f(\bar{\boldsymbol{z}}^{(l)}) \|^2 + \frac{4(1 - p_k)(\sigma^2 + \zeta^2)}{p_k} \eta^2,$$

*where $\bar{\boldsymbol{z}}^{(t)} := \frac{1}{k} \sum_{i=1}^{k} \boldsymbol{z}_i^{(t)}$ and $\Xi^{(t)} := \frac{1}{k} \sum_{i=1}^{k} \mathbb{E} \| \boldsymbol{z}_i^{(t)} - \bar{\boldsymbol{z}}^{(t)} \|^2$.*

*Proof.* From Lemma 5, we get

$$\Xi^{(t+1)} \leq \frac{6(1 - p_k) \eta^2}{p_k} \sum_{l=0}^{t} (1 - \frac{p_k}{4})^{t-l} \mathbb{E} \| \nabla f(\bar{\boldsymbol{z}}^{(l)}) \|^2 + (1 - p_k)(\sigma^2 + \zeta^2) \eta^2 \sum_{l=0}^{t} (1 - \frac{p_k}{4})^{t-l}$$

$$\leq \frac{6(1 - p_k) \eta^2}{p_k} \sum_{l=0}^{t} (1 - \frac{p_k}{4})^{t-l} \mathbb{E} \| \nabla f(\bar{\boldsymbol{z}}^{(l)}) \|^2 + \frac{4(1 - p_k)(\sigma^2 + \zeta^2)}{p_k} \eta^2.$$

By summing up the above inequality from $t = 0$ to $T - 1$, we obtain

$$
\begin{aligned}
\frac{1}{T+1}\sum_{t=1}^{T}\Xi^{(t)} &\leq \frac{6(1-p_k)\eta^2}{p_k(T+1)}\sum_{t=1}^{T}\sum_{l=0}^{t-1}(1-\frac{p_k}{4})^{t-l}\mathbb{E}\|\nabla f(\bar{z}^{(l)})\|^2 + \frac{4(1-p_k)(\sigma^2+\zeta^2)}{p_k}\eta^2 \\
&= \frac{6(1-p_k)\eta^2}{p_k(T+1)}\sum_{l=0}^{T-1}\mathbb{E}\|\nabla f(\bar{z}^{(l)})\|^2\sum_{t=l+1}^{T}(1-\frac{p_k}{4})^{t-l} + \frac{4(1-p_k)(\sigma^2+\zeta^2)}{p_k}\eta^2 \\
&= \frac{6(1-p_k)\eta^2}{p_k(T+1)}\sum_{l=0}^{T-1}\mathbb{E}\|\nabla f(\bar{z}^{(l)})\|^2\sum_{t'=1}^{T-l}(1-\frac{p_k}{4})^{t'} + \frac{4(1-p_k)(\sigma^2+\zeta^2)}{p_k}\eta^2 \\
&\leq \frac{24(1-p_k)\eta^2}{p_k^2(T+1)}\sum_{l=0}^{T-1}\mathbb{E}\|\nabla f(\bar{z}^{(l)})\|^2 + \frac{4(1-p_k)(\sigma^2+\zeta^2)}{p_k}\eta^2.
\end{aligned}
$$

Using $\Xi^{(0)} = 0$ and $\mathbb{E}\|\nabla f(\bar{z}^{(T)})\|^2 \geq 0$, we obtain the desired result. $\qquad\square$

**Lemma 7.** *Suppose that Assumptions 2, 3, 4, and 6 hold, and $\{x_i^{(0)}\}_{i=1}^{n}$ is initialized with the same parameter $\bar{x}^{(0)}$. If the step size satisfies $\eta \leq \frac{p_k}{\sqrt{192L}}$, then it holds that*

$$
\begin{aligned}
&\frac{1}{T+1}\sum_{t=0}^{T}\mathbb{E}\|\nabla f(\bar{x}_{active}^{(t)})\|^2 \\
&\leq \frac{8(f(\bar{x}^{(0)}) - f^\star)}{(T+1)\eta} + \frac{32L^2(\sigma^2+\zeta^2)(1-p_k)}{p_k}\eta^2 + \frac{4L}{k}\left(\sigma^2 + \left(1 - \frac{k-1}{n-1}\right)\zeta^2\right)\eta,
\end{aligned}
\tag{11}
$$

*where $\bar{x}_{active}^{(t)} := \frac{1}{k}\sum_{v_i \in V_{active}^{(t)}} x_i^{(t)}$.*

*Proof.* Rearranging Lemma 4, we obtain

$$
\mathbb{E}\|\nabla f(\bar{z}^{(t)})\|^2 \leq \frac{4(\mathbb{E}f(\bar{z}^{(t)}) - \mathbb{E}f(\bar{z}^{(t+1)}))}{\eta} + 4L^2\Xi^{(t)} + \frac{2L\sigma^2}{k}\eta + \frac{2L\zeta^2}{k}\left(1 - \frac{k-1}{n-1}\right)\eta.
$$

Using Lemma 6, we get

$$
\begin{aligned}
&\frac{1}{T+1}\sum_{t=0}^{T}\mathbb{E}\|\nabla f(\bar{z}^{(t)})\|^2 \\
&\leq \frac{4(f(\bar{z}^{(0)}) - f^\star)}{(T+1)\eta} + \frac{4L^2}{T+1}\Xi^{(t)} + \frac{2L\sigma^2}{k}\eta + \frac{2L\zeta^2}{k}\left(1 - \frac{k-1}{n-1}\right)\eta \\
&\leq \frac{4(f(\bar{z}^{(0)}) - f^\star)}{(T+1)\eta} + \frac{96L^2(1-p_k)}{p_k^2(T+1)}\eta^2\sum_{t=0}^{T}\mathbb{E}\|\nabla f(\bar{z}^{(l)})\|^2 + \frac{16L^2(1-p_k)(\sigma^2+\zeta^2)}{p_k}\eta^2 \\
&\quad + \frac{2L\sigma^2}{k}\eta + \frac{2L\zeta^2}{k}\left(1 - \frac{k-1}{n-1}\right)\eta.
\end{aligned}
$$

Using $\eta \leq \frac{p_k}{\sqrt{192L}}$ and $\bar{x}_{active}^{(t)} = \bar{z}^{(t)}$ (see Eq. (10)), we obtain the desired result. $\qquad\square$

**Lemma 8.** *Suppose that Assumptions 2, 3, 4, and 6 hold, and $\{x_i^{(0)}\}_{i=1}^{n}$ is initialized with the same parameter $\bar{x}^{(0)}$. There exists a step size $\eta$ that satisfies*

$$
\begin{aligned}
&\frac{1}{T+1}\sum_{t=0}^{T}\mathbb{E}\|\nabla f(\bar{x}_{active}^{(t)})\|^2 \\
&\leq \mathcal{O}\left(\sqrt{\frac{Lr_0(\sigma^2 + (1-\frac{k-1}{n-1})\zeta^2)}{kT}} + \left(\frac{L^2r_0^2(\sigma^2+\zeta^2)(1-p_k)}{T^2p_k}\right)^{\frac{1}{3}} + \frac{Lr_0}{Tp_k}\right),
\end{aligned}
\tag{12}
$$

*where $\bar{x}_{active}^{(t)} := \frac{1}{k}\sum_{v_i \in V_{active}^{(t)}} x_i^{(t)}$ and $r_0 := f(\bar{x}^{(0)}) - f^\star$.*

*Proof.* Choosing the step size as follows:

$$\eta = \min\left\{\sqrt{\frac{2kr_0}{L(T+1)(\sigma^2+\zeta^2)}}, \left(\frac{r_0 p_k}{4L^2(T+1)(1-p_k)(\sigma^2+\zeta^2)}\right)^{\frac{1}{3}}, \frac{p_k}{\sqrt{192L}}\right\},$$

we have three cases:

- When $\eta = \sqrt{\frac{2kr_0}{L(T+1)(\sigma^2+(1-\frac{k-1}{n-1})\zeta^2)}}$, $\frac{1}{T+1}\sum_{t=0}^{T}\mathbb{E}\|\nabla f(\bar{\boldsymbol{x}}_{\text{active}}^{(t)})\|^2$ is bounded from above by

$$\sqrt{\frac{128Lr_0\left(\sigma^2+(1-\frac{k-1}{n-1})\zeta^2\right)}{k(T+1)}} + \left(\frac{2048L^2r_0^2(1-p_k)(\sigma^2+\zeta^2)}{(T+1)^2 p_k}\right)^{\frac{1}{3}}.$$

- When $\eta = \left(\frac{r_0 p_k}{4L^2(T+1)(1-p_k)(\sigma^2+\zeta^2)}\right)^{\frac{1}{3}}$, $\frac{1}{T+1}\sum_{t=0}^{T}\mathbb{E}\|\nabla f(\bar{\boldsymbol{x}}_{\text{active}}^{(t)})\|^2$ is bounded from above by

$$\sqrt{\frac{32Lr_0\left(\sigma^2+(1-\frac{k-1}{n-1})\zeta^2\right)}{k(T+1)}} + \left(\frac{16384L^2r_0^2(1-p_k)(\sigma^2+\zeta^2)}{(T+1)^2 p_k}\right)^{\frac{1}{3}}.$$

- When $\eta = \frac{p_k}{\sqrt{192L}}$, $\frac{1}{T+1}\sum_{t=0}^{T}\mathbb{E}\|\nabla f(\bar{\boldsymbol{x}}_{\text{active}}^{(t)})\|^2$ is bounded from above by

$$\frac{\sqrt{12288}Lr_0}{(T+1)p_k} + \sqrt{\frac{32Lr_0\left(\sigma^2+(1-\frac{k-1}{n-1})\zeta^2\right)}{k(T+1)}} + \left(\frac{2048L^2r_0^2(1-p_k)(\sigma^2+\zeta^2)}{(T+1)^2 p_k}\right)^{\frac{1}{3}}.$$

□

**Lemma 9.** *Suppose that Assumptions 2, 3, 4, and 6 hold, and $\{\boldsymbol{x}_i^{(0)}\}_{i=1}^n$ is initialized with the same parameter $\bar{\boldsymbol{x}}^{(0)}$. Then, if the active nodes are connected by a ring, i.e., $p_k = \Omega(k^{-2})$, there exists a step size $\eta > 0$ and the number of active nodes $k \in \{1, 2, \cdots, n\}$ that satisfies*

$$\frac{1}{T+1}\sum_{t=0}^{T}\mathbb{E}\|\nabla f(\bar{\boldsymbol{x}}_{active}^{(t)})\|^2$$

$$\leq \mathcal{O}\left(\sqrt{\frac{Lr_0\sigma^2}{nT}} + \left(\frac{Lr_0(\sigma^2+\zeta^2)^{\frac{3}{4}}}{T}\right)^{\frac{4}{7}} + \left(\frac{Lr_0(\sigma^2+\zeta^2)^{\frac{2}{5}}}{T}\right)^{\frac{5}{7}} + \frac{Lr_0}{T}\right),$$

*where $\bar{\boldsymbol{x}}_{active}^{(t)} := \frac{1}{k}\sum_{v_i \in V_{active}^{(t)}}\boldsymbol{x}_i^{(t)}$ and $r_0 := f(\bar{\boldsymbol{x}}^{(0)}) - f^\star$.*

*Proof.* From Lemma 8, we obtain

$$\frac{1}{T+1}\sum_{t=0}^{T}\mathbb{E}\|\nabla f(\bar{\boldsymbol{x}}_{\text{active}}^{(t)})\|^2$$

$$\leq \mathcal{O}\left(\sqrt{\frac{Lr_0(\sigma^2+(1-\frac{k-1}{n-1})\zeta^2)}{kT}} + \left(\frac{L^2r_0^2(\sigma^2+\zeta^2)k^2}{T^2}\right)^{\frac{1}{3}} + \frac{Lr_0 k^2}{T}\right). \quad (13)$$

For simplicity, let $A$, $B$, and $C$ denote as follows:

$$A = \frac{Lr_0(\sigma^2+\zeta^2)}{T}, \qquad B = \left(\frac{L^2r_0^2(\sigma^2+\zeta^2)}{T^2}\right)^{\frac{1}{3}}, \qquad C = \frac{Lr_0}{T}.$$

Using these notations, we obtain

$$\frac{1}{T+1}\sum_{t=0}^{T}\mathbb{E}\|\nabla f(\bar{\boldsymbol{x}}_{\text{active}}^{(t)})\|^2 \leq \mathcal{O}\left(\sqrt{\frac{A}{k}}+Bk^{\frac{2}{3}}+Ck^2\right). \tag{14}$$

We choose $k$ as follows:

$$k = \max\left\{1, \min\left\{\left\lceil\left(\frac{A^3}{B^6}\right)^{\frac{1}{7}}\right\rceil, n\right\}\right\}.$$

Note that it holds that $k \in \{1, 2, 3, \cdots, n\}$.

When $r_0(\sigma^2 + \zeta^2) = 0$, it holds that $k = 1$, $A = 0$, and $B = 0$. In this case, $\frac{1}{T+1}\sum_{t=0}^{T}\mathbb{E}\|\nabla f(\bar{\boldsymbol{x}}_{\text{active}}^{(t)})\|^2$ is bounded from above by

$$\mathcal{O}\left(\frac{Lr_0}{T}\right).$$

When $r_0(\sigma^2 + \zeta^2) > 0$, it holds that

$$k = \min\left\{\left\lceil\left(\frac{A^3}{B^6}\right)^{\frac{1}{7}}\right\rceil, n\right\}.$$

In this case, we have three cases:

- When $k = \left\lceil\left(\frac{A^3}{B^6}\right)^{\frac{1}{7}}\right\rceil$, we have

$$\sqrt{\frac{A}{k}} = \mathcal{O}\left(A^{\frac{2}{7}}B^{\frac{3}{7}}\right), \quad Bk^{\frac{2}{3}} = \mathcal{O}\left(A^{\frac{2}{7}}B^{\frac{3}{7}}\right), \quad Ck^2 = \mathcal{O}\left(CA^{\frac{6}{7}}B^{\frac{-12}{7}}\right).$$

  By using the above inequalities, $\frac{1}{T+1}\sum_{t=0}^{T}\mathbb{E}\|\nabla f(\bar{\boldsymbol{x}}_{\text{active}}^{(t)})\|^2$ is bounded from above by

$$\mathcal{O}\left(A^{\frac{2}{7}}B^{\frac{3}{7}}+A^{\frac{6}{7}}B^{\frac{-12}{7}}C\right).$$

- When $k = n$, we have

$$\sqrt{\frac{Lr_0(\sigma^2 + (1-\frac{k-1}{n-1})\zeta^2)}{kT}} = \sqrt{\frac{Lr_0\sigma^2}{nT}}, \quad Bk^{\frac{2}{3}} \leq \mathcal{O}\left(A^{\frac{2}{7}}B^{\frac{3}{7}}\right), \quad Ck^2 \leq \mathcal{O}\left(CA^{\frac{6}{7}}B^{\frac{-12}{7}}\right),$$

  where we use $k \leq \left\lceil\left(\frac{A^3}{B^6}\right)^{\frac{1}{7}}\right\rceil$. By using the above inequalities, $\frac{1}{T+1}\sum_{t=0}^{T}\mathbb{E}\|\nabla f(\bar{\boldsymbol{x}}_{\text{active}}^{(t)})\|^2$ is bounded from above by

$$\mathcal{O}\left(\sqrt{\frac{Lr_0\sigma^2}{nT}}+A^{\frac{2}{7}}B^{\frac{3}{7}}+A^{\frac{6}{7}}B^{\frac{-12}{7}}C\right).$$

By summarizing the above inequalities, we obtain the desired results. □

**Lemma 10.** *Suppose that Assumptions 2, 3, 4, and 6 hold, and $\{\boldsymbol{x}_i^{(0)}\}_{i=1}^n$ is initialized with the same parameter $\bar{\boldsymbol{x}}^{(0)}$. Then, if the active nodes are connected by an exponential graph, i.e., $p_k = \Omega(\log_2^{-1} k)$, there exists a step size $\eta > 0$ and the number of active nodes $k \in \{1, 2, \cdots, n\}$ that satisfies*

$$\frac{1}{T+1}\sum_{t=0}^{T}\mathbb{E}\|\nabla f(\bar{\boldsymbol{x}}_{active}^{(t)})\|^2$$

$$\mathcal{O}\left(\sqrt{\frac{Lr_0\sigma^2}{nT}}+\left(\frac{L^2r_0^2(\sigma^2+\zeta^2)}{T^2}\log_2\left(\frac{T(\sigma^2+\zeta^2)}{Lr_0}\right)^{\frac{1}{3}}\right)^{\frac{1}{3}}+\frac{Lr_0}{T}\log_2\left(\frac{T(\sigma^2+\zeta^2)}{Lr_0}\right)+\frac{Lr_0}{T}\right),$$

*where $\bar{\boldsymbol{x}}_{active}^{(t)} := \frac{1}{k}\sum_{v_i \in V_{active}^{(t)}}\boldsymbol{x}_i^{(t)}$ and $r_0 := f(\bar{\boldsymbol{x}}^{(0)}) - f^\star$.*

*Proof.* From Lemma 8, we obtain

$$\frac{1}{T+1}\sum_{t=0}^{T}\mathbb{E}\|\nabla f(\bar{\boldsymbol{x}}_{\text{active}}^{(t)})\|^2$$

$$\leq \mathcal{O}\left(\sqrt{\frac{Lr_0(\sigma^2+(1-\frac{k-1}{n-1})\zeta^2)}{kT}}+\left(\frac{L^2r_0^2(\sigma^2+\zeta^2)\log_2(k)}{T^2}\right)^{\frac{1}{3}}+\frac{Lr_0\log_2(k)}{T}\right).$$

For simplicity, let $A$, $B$, and $C$ denote as follows:

$$A = \frac{Lr_0(\sigma^2+\zeta^2)}{T}, \qquad B = \left(\frac{L^2r_0^2(\sigma^2+\zeta^2)}{T^2}\right)^{\frac{1}{3}}, \qquad C = \frac{Lr_0}{T}.$$

Using these notations, we obtain

$$\frac{1}{T+1}\sum_{t=0}^{T}\mathbb{E}\|\nabla f(\bar{\boldsymbol{x}}_{\text{active}}^{(t)})\|^2 \leq \mathcal{O}\left(\sqrt{\frac{A}{k}}+B\log_2^{\frac{1}{3}}(k)+C\log_2(k)\right).$$

We choose $k$ as follows:

$$k = \max\left\{1, \min\left\{\left\lceil\frac{A}{B^2}\right\rceil, \left\lceil\frac{A}{C^2}\right\rceil, n\right\}\right\}.$$

When $r_0(\sigma^2+\zeta^2) = 0$, it holds that $k = 1$, $A = 0$, and $B = 0$. In this case, $\frac{1}{T+1}\sum_{t=0}^{T}\mathbb{E}\|\nabla f(\bar{\boldsymbol{x}}_{\text{active}}^{(t)})\|^2$ is bounded from above by

$$\mathcal{O}\left(\frac{Lr_0}{T}\right).$$

When $r_0(\sigma^2+\zeta^2) > 0$, it holds that

$$k = \min\left\{\left\lceil\frac{A}{B^2}\right\rceil, \left\lceil\frac{A}{C^2}\right\rceil, n\right\}.$$

In this case, we have three cases:

- When $k = \left\lceil\frac{A}{B^2}\right\rceil$, $\frac{1}{T+1}\sum_{t=0}^{T}\mathbb{E}\|\nabla f(\bar{\boldsymbol{x}}_{\text{active}}^{(t)})\|^2$ is bounded from above by

$$\mathcal{O}\left(B\log_2^{\frac{1}{3}}\left(\frac{A}{B^2}\right)+C\log_2\left(\frac{A}{C^2}\right)\right).$$

- When $k = \left\lceil\frac{A}{C^2}\right\rceil$, $\frac{1}{T+1}\sum_{t=0}^{T}\mathbb{E}\|\nabla f(\bar{\boldsymbol{x}}_{\text{active}}^{(t)})\|^2$ is bounded from above by

$$\mathcal{O}\left(B\log_2^{\frac{1}{3}}\left(\frac{A}{B^2}\right)+C\log_2\left(\frac{A}{C^2}\right)\right).$$

- When $k = n$, $\frac{1}{T+1}\sum_{t=0}^{T}\mathbb{E}\|\nabla f(\bar{\boldsymbol{x}}_{\text{active}}^{(t)})\|^2$ is bounded from above by

$$\mathcal{O}\left(\sqrt{\frac{Lr_0\sigma^2}{nT}}+B\log_2^{\frac{1}{3}}\left(\frac{A}{B^2}\right)+C\log_2\left(\frac{A}{C^2}\right)\right).$$

By summarizing the above inequalities, we obtain the desired result. □

## C    PROOF OF THEOREM 3

**Lemma 1.** *For any $k^\star < n$, there exists $k \in \{1, 2, 4, 8, \cdots, 2^{\lfloor \log_2(n+1) \rfloor - 1}\}$ that satisfies $\frac{k^\star}{4} < k \leq k^\star$. Furthermore, it holds that $\sum_{i=0}^{\lfloor \log_2(n+1) \rfloor - 1} 2^i \leq n$.*

*Proof.* We define $\mathcal{K} := \{1, 2, 4, 8, \cdots, 2^{\lfloor \log_2(n+1) \rfloor - 1}\}$. We have two cases:

- When $1 \leq k^\star < 2^{\lfloor \log_2(n+1) \rfloor}$, there exists $k \in \mathcal{K}$ that satisfies $k \leq k^\star < 2k$.

- When $2^{\lfloor \log_2(n+1) \rfloor} \leq k^\star < n$, it holds that $k < k^\star$ for all $k \in \mathcal{K}$. Then, we have
$$k^\star < n < 4 \times 2^{\lfloor \log_2(n+1) \rfloor - 1}.$$
  Thus, there exists $k \in \mathcal{K}$ that satisfies $k < k^\star < 4k$.

By combining the above two cases, we obtain the desired result.    $\square$

**Lemma 11.** *Suppose that Assumptions 1, 2, 3, 4, and 6 hold. Let $\{\{\boldsymbol{x}_{k,i}^{(t)}\}_{v_i \in V_{active}^{(t)}}\}_{t=0}^T$ denote the parameters of active nodes generated by Alg. 3 when the number of active nodes is set to $k$ and define $\mathcal{K} := \{1, 2, 2^2, 2^3, \cdots, 2^{\lfloor \log_2(n+1) \rfloor - 1}, n\}$. Then, suppose that the parameters are initialized with the same parameter $\bar{\boldsymbol{x}}^{(0)}$.*

***Ring:*** *If the active nodes are connected by a ring, i.e., $p_k = \Omega(k^{-2})$, there exists $\eta$ such that $\min_{k \in \mathcal{K}} \left( \frac{1}{T+1} \sum_{t=0}^T \mathbb{E}\|\nabla f(\bar{\boldsymbol{x}}_{active,k}^{(t)})\|^2 \right)$ is bounded from above by*

$$\mathcal{O}\left( \sqrt{\frac{Lr_0\sigma^2}{nT}} + \left( \frac{Lr_0(\sigma^2 + \zeta^2)^{\frac{3}{4}}}{T} \right)^{\frac{4}{7}} + \left( \frac{Lr_0(\sigma^2 + \zeta^2)^{\frac{2}{3}}}{T} \right)^{\frac{3}{5}} + \frac{Lr_0}{T} \right),$$

*where $r_0 := f(\bar{\boldsymbol{x}}_{active,k}^{(0)}) - f^\star$ and $\bar{\boldsymbol{x}}_{active,k}^{(t)} := \frac{1}{k} \sum_{v_i \in V_{active}^{(t)}} \boldsymbol{x}_{k,i}$.*

***Exp. Graph:*** *If the active nodes are connected by an exponential graph, i.e., $p_k = \Omega(\log_2^{-1} k)$, there exists $\eta$ such that $\min_{k \in \mathcal{K}} \left( \frac{1}{T+1} \sum_{t=0}^T \mathbb{E}\|\nabla f(\bar{\boldsymbol{x}}_{active,k}^{(t)})\|^2 \right)$ is bounded from above by*

$$\mathcal{O}\left( \sqrt{\frac{Lr_0\sigma^2}{nT}} + \left( \frac{L^2 r_0^2(\sigma^2 + \zeta^2)}{T^2} \log_2\left( \frac{T(\sigma^2 + \zeta^2)}{Lr_0} \right)^{\frac{1}{3}} \right)^{\frac{1}{3}} + \frac{Lr_0}{T} \log_2\left( \frac{T(\sigma^2 + \zeta^2)}{Lr_0} \right) + \frac{Lr_0}{T} \right).$$

*Proof.* By combining Lemmas 1, 9, and 10, we obtain the statement.    $\square$

# D CONVERGENCE RATE OF DECENTRALIZED SGD WITH VARIOUS TOPOLOGIES

Table 2: Convergence rates of DSGD over various topologies.

| Topology | Convergence Rate |
|---|---|
| Ring (Nedić et al., 2018) | $\mathcal{O}\left(\sqrt{\frac{Lr_0\sigma^2}{nT}} + \left(\frac{L^2r_0^2n^2(\sigma^2+n^2\zeta^2)}{T^2}\left(1-\frac{1}{n^2}\right)\right)^{\frac{1}{3}} + \frac{Lr_0n^2}{T}\right)$ |
| Torus (Nedić et al., 2018) | $\mathcal{O}\left(\sqrt{\frac{Lr_0\sigma^2}{nT}} + \left(\frac{L^2r_0^2n(\sigma^2+n\zeta^2)}{T^2}\left(1-\frac{1}{n}\right)\right)^{\frac{1}{3}} + \frac{Lr_0n}{T}\right)$ |
| Exponential Graph (Ying et al., 2021) | $\mathcal{O}\left(\sqrt{\frac{Lr_0\sigma^2}{nT}} + \left(\frac{L^2r_0^2\log_3(n)(\sigma^2+\log_2(n)\zeta^2)}{T^2}\left(1-\frac{1}{\log_2(n)}\right)\right)^{\frac{1}{3}} + \frac{Lr_0\log_2(n)}{T}\right)$ |
| Base-2 Graph (Takezawa et al., 2023b) | $\mathcal{O}\left(\sqrt{\frac{Lr_0\sigma^2}{nT}} + \left(\frac{L^2r_0^2\log_2(n)(\sigma^2+\log_2(n)\zeta^2)}{T^2}\right)^{\frac{1}{3}} + \frac{Lr_0\log_2(n)}{T}\right)$ |

# E CONVERGENCE RATE OF DECENTRALIZED SGD WITH CLIENT SAMPLING

**Proposition 2** (Liu et al. (2022)). *Suppose that Assumptions 1, 2, 3, 4, and 5 hold. Let $k$ be the number of active nodes and $\{\{\boldsymbol{x}_i^{(t)}\}_{i=1}^n\}_{t=0}^T$ denote the parameters generated by Decentralized SGD with client sampling. Then, there exists the step size $\eta$ that satisfies:*

$$\frac{1}{T+1}\sum_{t=0}^T \mathbb{E}\|\nabla f(\bar{\boldsymbol{x}}^{(t)})\|^2 \leq \mathcal{O}\left(\sqrt{\frac{Lr_0(\sigma^2+(1-\frac{k-1}{n-1})\zeta^2)}{kT}} + \left(\frac{n}{k}\frac{Lr_0(\sigma\sqrt{p_n}+\zeta^2)}{Tp_n}\right)^{\frac{2}{3}} + \frac{Lr_0}{Tp_n}\right),$$

*where $r_0 := f(\bar{\boldsymbol{x}}^{(0)}) - f^\star$ and $\bar{\boldsymbol{x}}^{(t)} := \frac{1}{n}\sum_{i=1}^n \boldsymbol{x}_i^{(t)}$.*

**Remark 1.** *The convergence rate in Proposition 2 consistently deteriorates as $k$ decreases.*

Unlike TELEPORTATION, the convergence rate shown in Proposition 2 depends on $p_n$, and the second and third terms degrade as $n$ increases.

# F ADDITIONAL EXPERIMENT

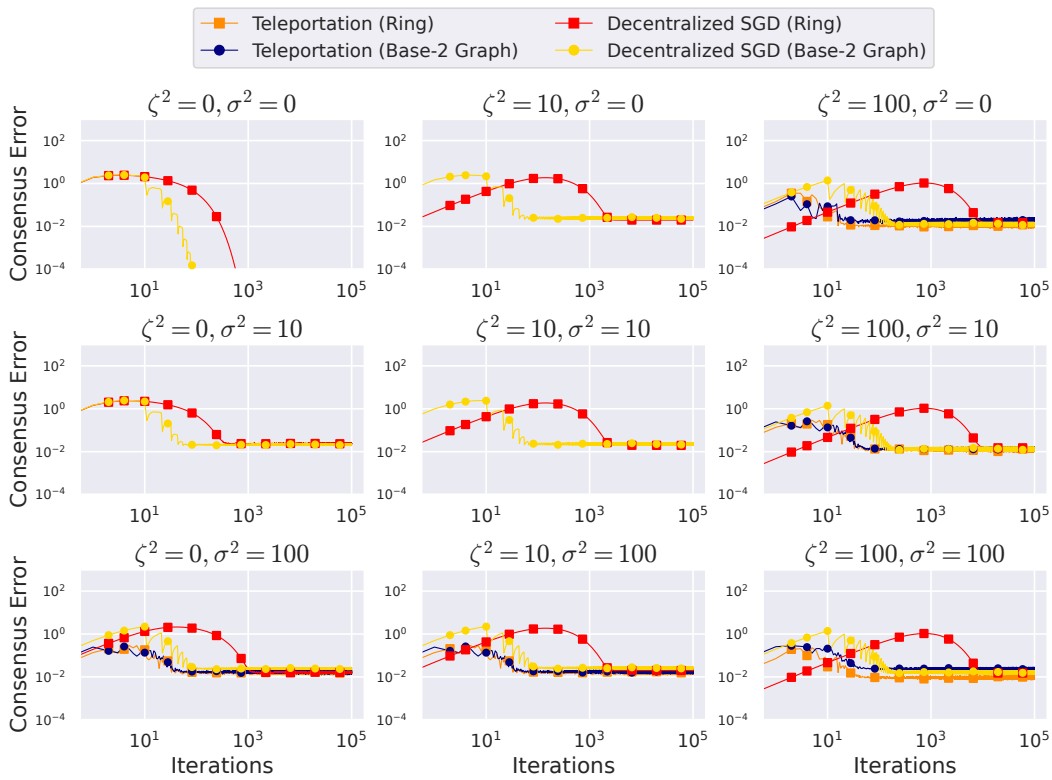

Figure 6: The behavior of consensus error $\frac{1}{k} \sum_{v_i \in V_{\text{active}}^{(t)}} \|\boldsymbol{x}_i^{(t)} - \bar{\boldsymbol{x}}_{\text{active}}^{(t)}\|^2$ for different stochastic noise $\sigma^2$ and heterogeneity $\zeta^2$. The experimental setup was the same as in Fig. 2. Note that when the optimal number of active nodes $k$ is one, we did not plot the consensus error. See Table 1.

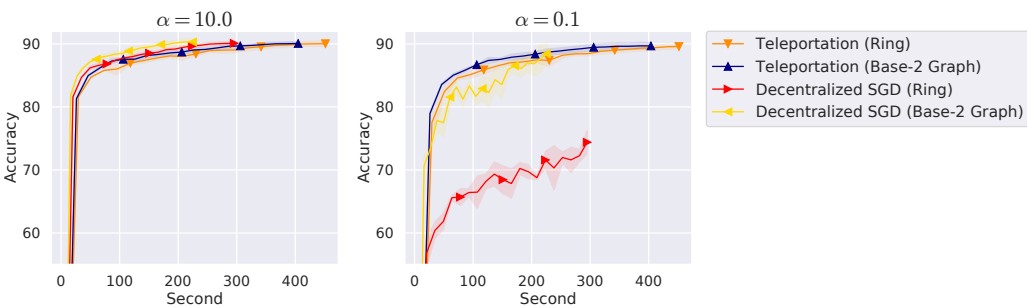

Figure 7: Test accuracy of TELEPORTATION and Decentralized SGD under heterogeneous networks with $\tau = 0$. The experimental setup was the same as in Fig. 4.

# G EXPERIMENTAL SETUP

Table 3: Experimental setups for Fig. 2.

| | |
|---|---|
| Step size | Grid search over $\{0.1, 0.075, 0.05, 0.025, 0.01, \cdots, 0.0001\}$ |
| Computational resources | AMD Epyc 7702 CPU or Intel Xeon Gold 6230 CPU |

Table 4: Experimental setups for Fashion MNIST in Fig. 3.

| | |
|---|---|
| Model | LeNet |
| Step size | Grid search over $\{0.1, 0.01, 0.001\}$ |
| Batch size | 32 |
| Momentum | 0.9 |
| Epoch | 200 |
| Computational resources | Titan $\times$ 8 |

Table 5: Experimental setups for CIFAR-10 in Fig. 3.

| | |
|---|---|
| Model | VGG |
| Step size | Grid search over $\{0.1, 0.01, 0.001\}$ |
| Scheduler | Cosine decay |
| Batch size | 32 |
| Momentum | 0.9 |
| Epoch | 500 |
| Computational resources | A6000 $\times$ 8 or RTX 3090 $\times$ 8 |

Table 6: Experimental setups for Figs. 4 and 7.

| | |
|---|---|
| Model | LeNet |
| Step size | Grid search over $\{0.1, 0.01, 0.001\}$ |
| Batch size | 32 |
| Momentum | 0.9 |
| Epoch | 200 |
| Computational resources | A6000 $\times$ 8 |

