# OpenReview forum: "Scalable Decentralized Learning with Teleportation"
_ICLR.cc/2025/Conference — ICLR 2025 Poster_

### Official Review · Reviewer_67Bz · 2024-10-30

**Soundness:** 3
**Presentation:** 3
**Contribution:** 2
**Rating:** 5
**Confidence:** 4

**Summary:**

The paper addresses degradation in decentralized learning arising from fixed communication patterns. Given the flexibility to select both a communication pattern and a subset of active nodes, the authors propose the TELEPORTATION algorithm. This algorithm activates k nodes per iteration and applies a specific topology to these nodes, facilitating adaptable communication.

The convergence analysis for TELEPORTATION follows the approach used by Koloskova (2020b), extending it to any number of active nodes k. Additionally, the authors provide an efficient method for tuning k, including determining optimal values for specific graph types, such as rings and exponential graphs, where they establish the theoretically best k and its associated convergence rate.

Experimental results demonstrate that TELEPORTATION consistently outperforms decentralized SGD across various settings and graph structures, reinforcing its potential advantages in decentralized learning.

**Strengths:**

The primary strength of this paper lies in its proof of Theorem 1, which effectively addresses a key limitation in previous analyses of similar algorithms. Prior approaches encountered challenges due to the disconnectivity of graphs under certain node activation conditions (e.g., when p=0), resulting in an infinite bound on the number of iterations required for convergence. Remarkably, this paper recovers the bound obtained by Kolosokova (2020b) under a generalized node selective activation scheme, which could prove potential for future research in this domain.

In addition to the theoretical insights, the paper is also notable for its clear presentation of contributions. The experimental section is both comprehensive and persuasive, further validating the authors' approach and supporting the potential utility of their bound.

**Weaknesses:**

Even though the distinction of TELEPORTATION from client sampling has been discussed, it still looks to me a follow-up variant in this category (McMahan 2017), and client sampling has been a heavily discussed approach. In the special case where any node is allowed to connect to any node, decentralized learning with client sampling is a very natural idea. Admittedly the proof of this paper could be something new, but the approach they proposed, at least in practice, might have been discussed/implemented allready. In particular, it is expected  when you select a fixed graph of size k, where its spectral gap is known for example in ring and exp. graph, the spectral gap does not show up in the bound. In this sense, "completely alleviate the convergence rate degradation" is not that interesting any more.

Minor: there are no black nodes in Figure 1.

**Questions:**

What are the Base-2 graph and what do you mean by superiority of this graph?

---

> ### Author Response · Authors · 2024-11-19
>
> We thank the reviewer for your comments.
>
> > Even though the distinction of TELEPORTATION from client sampling has been discussed, it still looks to me a follow-up variant in this category (McMahan 2017) [...] "completely alleviate the convergence rate degradation" is not that interesting any more.
>
> **We believe that the similarity between TELEPORTATION and client sampling does not mean that our paper is less novel.**
> Mitigating the degradation caused by large $n$ is an essential topic in decentralized learning, and our paper successfully proposed a method to solve this degradation.
>
> Scaling decentralized learning to a large number of nodes is a challenging and important research topic, since the convergence rate of DSGD degrades when $n$ is significantly large.
> Existing papers have attempted to alleviate this problem by proposing topologies with a large spectral gap [1,2,3].
> However, no existing work can completely eliminate the convergence rate degradation caused by large $n$ (see Table 2 in Sec. D).
> It has been an open question of how to eliminate this degradation completely.
>
> In this study, we solved this issue by activating only the appropriate number of nodes instead of designing a graph with a large spectral gap.
> Then, we found that TELEPORTATION can **completely** eliminate this degradation without sacrificing the linear speedup property $\mathcal{O}(1 / \sqrt{n T})$.
> We believe that it is surprising that such a simple idea can totally alleviate this degradation.
>
> As the reviewer mentioned, client sampling is widely studied in the federated learning literature to reduce the communication costs between the central server and nodes.
> However, it is novel to use the idea of activating only a few nodes to mitigate the degradation caused by large $n$ in decentralized learning,
> and it is not trivial that this idea can totally eliminate the degradation caused by $n$.
> We believe that the fact that the proposed method is similar to client sampling does not diminish our contribution.
>
>
> > What are the Base-2 graph and what do you mean by superiority of this graph?
>
> Base-2 Graph is a topology that is proposed to alleviate the convergence rate degradation caused by large $n$, and it can achieve a reasonable balance between the convergence rate and communication efficiency.
> [1] demonstrated that Base-2 Graph enables DSGD to more successfully reconcile accuracy and communication efficiency than the other existing topologies in their experiments.
> Thus, we compared TELEPORTATION and DSGD with Base-2 Graph in our experiments.
>
>
> ## Reference
>
> [1] Takezawa et. al., Beyond Exponential Graph: Communication-efficient Topologies for Decentralized Learning via Finite-time Convergence, In NeurIPS 2023
>
> [2] Ying et. al., Exponential Graph is Provably Efficient for Decentralized Deep Training, In NeurIPS 2021
>
> [3] Ding et. al., . DSGD-CECA: Decentralized SGD with Communication-optimal Exact Consensus Algorithm, In ICML 2023

---

> ### Comment · Reviewer_67Bz · 2024-11-24
>
> I appreciate the contributions of the paper; however, I feel that the presentation may overstate the significance of the results. The analysis by Koloskova et al. establishes a connection between the convergence rate and parameters $p$ and $n$, showing that the required convergence time decreases as $n$ increases and increases as $p$ decreases. In scenarios where
> $p$ depends on $𝑛$, as seen in all the cases presented in Table 2 of this paper, their bounds have the claimed degradation in terms of $n$.
>
> In the context where $p$ depends on $n$, denoted here as $p_n$, the primary contribution of the present paper is to replace $p_n$ in Koloskova et al.'s bounds with $p_k$. The authors then claim to have completely removed the degradation with respect to $n$. However, this claim appears to be somewhat overstated for the following reasons:
> 1. Dependency of $p_k$ on $n$: While $p_k$ is introduced as distinct from $p_n$, it is not entirely independent of $n$. For instance, in lines 277-282 regarding the Exp. Graph, $k$ is defined as $\max (1,n, \dots )$. Consequently, $p_k$ inherits a dependency on $n$ through $k$.
> 2. Addition of a new parameter: Even if $p_k$ does not depend on $n$, the introduction of $p_k$ adds an additional parameter to the analysis.. Improvements of this nature could also be achieved using other techniques (e.g. client activation, graph sparsification) that similarly parameterize $p$ with a variable independent of $n$.
> 3. Limited impact in certain cases: The paper does not demonstrate any improvement in cases where the spectral gap does not scale with $n$.

---

> ### Author Response · Authors · 2024-11-25
>
> We thank the reviewer for clarifying the concerns.
>
>
> We suspect the reviewer feels that our claim that TELEPORTATION can "completely" mitigate the degradation caused by $n$ is overstated.
> For example, we could remove this statement and modify this statement to be more accurate by claiming that “the convergence rate of TELEPORTATION consistently improves convergence rate as $n$ increases.”
>
> Would this modification address the reviewer's concerns?
> If the reviewer's concerns are still unresolved, please let us know. We will gladly address your concerns.
>
> See below for more detailed comments.
>
> > While $p_k$ is introduced as distinct from, it is not entirely independent of $n$. For instance, in lines 277-282 regarding the Exp. Graph, $p_k$ is defined as $\max (1, n, \dots)$. Consequently, $p_k$ inherits a dependency on $n$ through $k$.
>
> We agree with the reviewer that $p_k$ is not entirely independent of $n$ since the choice of $k$ depends on $n$.
> However, the dependence of $n$ is no longer harmful in TELEPORTATION, as shown in Theorem 2, where the convergence rate of TELEPORTATION consistently improves as $n$ increases.
> For this reason, we claimed that TELEPORTATION can completely eliminate the convergence rate degradation caused by large $n$ in the current manuscript, but if the reviewer feels that this claim is an overstatement, we would be happy to revise it as mentioned above.
>
> > Even if $p_k$ does not depend on $n$, the introduction of $p_k$ adds an additional parameter to the analysis.. Improvements of this nature could also be achieved using other techniques (e.g. client activation, graph sparsification) that similarly parameterize $p$ with a variable independent of $n$.
>
> As the reviewer mentioned, we need to make $p$ independent of $n$ to alleviate the convergence rate degradation caused by large $n$.
>
> However, we would like to emphasize again that no existing paper has succeeded in eliminating the degradation associated with large $n$ (see Table 2 in Sec. D), and our paper is the first to propose a decentralized learning method whose convergence rate can consistently improve as $n$ increases.
>
> The reviewer mentioned that other techniques might achieve results similar to TELEPORTATION, but we believe the possibility of unpublished alternative methods does not diminish our contribution.
>
> > The paper does not demonstrate any improvement in cases where the spectral gap does not scale with $n$.
>
> In decentralized learning, it is essential to reconcile the communication efficiency and convergence rate, e.g., $p_n$.
> As we discussed in our paper, no existing paper has succeeded in proposing a topology that has a spectral gap that does not decrease as $n$ increases without sacrificing the communication efficiency (see Table 2 in Sec. D).
> TELEPORTATION is the first method that can eliminate the convergence rate degradation caused by large $n$ without sacrificing the communication efficiency.

---

> > ### Author Response · Authors · 2024-11-29
> >
> > We appreciate your efforts in reviewing our paper.
> > The rebuttal deadline is approaching. We would like to revise our manuscript to ensure that our claims are accurately presented and that readers do not feel that our claim is overstated. We would greatly appreciate it if you could kindly provide feedback at your convenience.
> > If there are any further questions, we would be happy to address them.

---

### Official Review · Reviewer_bkGn · 2024-11-04

**Soundness:** 3
**Presentation:** 3
**Contribution:** 2
**Rating:** 6
**Confidence:** 3

**Summary:**

This paper introduces a method called TELEPORTATION aimed at improving decentralized learning, particularly decentralized stochastic gradient descent (SGD).

TELEPORTATION is have a better dependence on the spectral gap of the communication graph in the convergence rate. So, the authors claim that their method can alleviate the convergence rate degradation while maintaining communication efficiency.

They also propose a method to optimize the number of active nodes so as to fasten the convergence rate.

The experiments suggest that TELEPORTATION outperforms decentralized SGD in both convergence speed and stability, especially when data distribution is uneven across nodes.

**Strengths:**

1. **Reduced Communication Overhead**:

The paper proposes a practical solution to reduce the communication costs in decentralized learning by activating fewer nodes. The methods developed to tune the hyperparameters would inspire future work.

2. **Clear Theoretical Analysis**:

The paper provides solid theoretical support for the claims, including convergence rate bounds and proofs for the proposed method.

3. The paper is mostly well-written and easy to follow.

**Weaknesses:**

1. **Lack of Clarity in Problem Formulation**:
   The paper could benefit from a clearer and more formal description of the problems posed by decentralized learning, particularly why more nodes lead to degraded performance. Without this clarification, it becomes difficult to fully appreciate the novelty and impact of TELEPORTATION.

2. **Loose Theoretical Guarantees**:

   Beyond Proposition 1, the paper could be more rigorous in establishing the tightness of the bounds used to justify the method's benefits. A deeper exploration of whether these bounds hold tightly in practical cases (e.g., through empirical validation or consider quadratic cases) would enhance confidence in the theoretical contributions.

3. **Limited Practical Applicability**:

   The assumption that any two nodes can communicate directly (as mentioned in the abstract and discussions) is overly idealized. In many real-world applications, such as wireless sensor networks or geographically distributed systems, this assumption may not hold.

**Questions:**

1. **Convergence and the Number of Nodes**:
   I am unsure why decentralized learning methods should necessarily converge more slowly as the number of nodes increases (line 53-line 60). In fact, in fully synchronized systems, having more nodes typically leads to a linear speedup in convergence. It would be helpful if the authors could clearly formulate why this slowdown happens in decentralized settings and explicitly highlight the limitations of existing methods. A more specific problem formulation, accompanied by examples illustrating why current approaches fail to address this, would make the contribution clearer. (e.g., using a table summarizing existing results).

2. **Discussion on Proposition 1**:

   The discussion in lines 140-148 seems to rely on the upper bound from Proposition 1. However, this argument feels weak, as the tightness of the upper bound is never discussed. If the bound is loose due to flaws in the analysis, the subsequent discussion would be rendered meaningless. Is there any way to demonstrate that the bound is tight? For instance, you could explore the behavior of a simple quadratic function to provide more concrete support for this bound.

3. **Explanation of Theoretical Improvements**:

   It appears that the theoretical improvement primarily stems from the dependence on $p_n$. Could the authors clarify the source of this improvement? Is it driven by a more refined analysis, or is it due to a novel aspect of the algorithm itself? Providing a brief but clear explanation of what underlies this theoretical gain would strengthen the paper.

4. **Unclear Impact of Hyperparameter Tuning**:

   The proposed hyperparameter-tuning method for selecting the number of active nodes adds another layer of complexity to the method. However, the impact of this additional tuning on overall training time and resource consumption is not fully explored. In practice, the benefits of communication reduction could be outweighed by the cost of hyperparameter search. A detailed examination of this trade-off would provide more practical insight.

---

> ### Author Response · Authors · 2024-11-19
> **Official Comment by Authors (1/2)**
>
> We thank the reviewer for our comments.
>
> > I am unsure why decentralized learning methods should necessarily converge more slowly as the number of nodes increases [...]
>
> We would like to briefly explain why the convergence rate of DSGD can be degraded as $n$ is substantially large.
>
> DSGD satisfies the following:
>
> \begin{align}
>     \bar{\mathbf{x}}^{(t+1)} = \bar{\mathbf{x}}^{(t)} - \frac{\eta}{n} \sum_{i=1}^n \nabla F_i (\mathbf{x}_i ; \xi_i^{(t)}),
> \end{align}
>
> where $\bar{\mathbf{x}} \coloneqq \frac{1}{n} \sum_{i=1}^n \mathbf{x}_i$ (see the proof of Lemma 11 in [1]).
> $\nabla F_i$ is calculated at different parameters $\mathbf{x}_i$, and the convergence rate degrades when $\mathbf{x}_i$ is far from $\bar{\mathbf{x}}$.
> In decentralized learning, each node communicates with a few nodes for communication efficiency.
> For instance, if we use a ring as the underlying topology, each node communicates with two neighboring nodes.
> Thus, $\mathbf{x}_i$ comes to be far from $\bar{\mathbf{x}}$ as $n$ increases, and the convergence rate of DSGD degrades when $n$ is substantially large.
>
> The following table summarizes the convergence rate of DSGD with various topologies.
> For all topology, $n$ appears in the numerator in the convergence rate, and the rate degrades when $n$ is substantially large.
> We show this table in Sec. D.
>
> | Topology | Convergence Rate |
> | -------- | -------- |
> | Ring     | $\mathcal{O} \left( \sqrt{\frac{L r_0 \sigma^2}{n T}} + \left( \frac{L^2 r_0^2 n^2 (\sigma^2 + n^2 \zeta^2)}{T^2} \left( 1 - \frac{1}{n^2} \right) \right)^\frac{1}{3} + \frac{L r_0 n^2}{T} \right)$    |
> | Torus | $\mathcal{O} \left( \sqrt{\frac{L r_0 \sigma^2}{n T}} + \left( \frac{L^2 r_0^2 n (\sigma^2 + n \zeta^2)}{T^2} \left( 1 - \frac{1}{n} \right)\right)^\frac{1}{3} + \frac{L r_0 n}{T} \right)$ |
> | Exponential Graph | $\mathcal{O} \left( \sqrt{\frac{L r_0 \sigma^2}{n T}} + \left( \frac{L^2 r_0^2 \log_3 (n) (\sigma^2 + \log_2 (n) \zeta^2)}{T^2} \left( 1 - \frac{1}{\log_2 (n)} \right) \right)^\frac{1}{3} + \frac{L r_0 \log_2 (n)}{T} \right)$ |
> | Base-2 Graph     | $\mathcal{O} \left( \sqrt{\frac{L r_0 \sigma^2}{n T}} + \left( \frac{L^2 r_0^2 \log_2 (n) (\sigma^2 + \log_2 (n) \zeta^2)}{T^2} \right)^\frac{1}{3}  + \frac{L r_0 \log_2 (n)}{T} \right)$ |
>
>
> > The discussion in lines 140-148 seems to rely on the upper bound from Proposition 1. However, this argument feels weak, as the tightness of the upper bound is never discussed. [...] Is there any way to demonstrate that the bound is tight? [...]
>
> Thank you for the comment.
> **[1] analyzed the lower bound of the convergence rate of SGD and showed that it is inevitable that the convergence rate of DSGD degrades when $n$ is substantially large.**
> We will add the following discussion in the revised manuscript.
>
> Specifically, Theorem 3 in [1] shows the lower bound of the convergence rate of DSGD when $f_i$ is $\mu$-strongly convex and $L$-smooth with $\mu = L=1$ and $\sigma=0$.
> Under these assumptions, [1] showed that it requires
> \begin{align}
>     \tilde{\Omega} (\frac{\zeta (1-p_n)}{\sqrt{\epsilon} p_n})
> \end{align}
> iterations to converge to accuracy $\epsilon$.
> See Theorem 3 in [1] for the precise statement.
> Thus, the convergence rate of DSGD must depend on $p_n$, and it is inevitable that the convergence rate of DSGD degrades when $n$ is substantially large since $p_n$ reaches zero as $n$ increases.
>
> > It appears that the theoretical improvement primarily stems from the dependence on $p_n$. Could the authors clarify the source of this improvement? Is it driven by a more refined analysis, or is it due to a novel aspect of the algorithm itself? [...]
>
> As we mentioned above, it is inevitable that the convergence rate of DSGD degrades when $n$ is substantially large.
> **Thus, the property that TELEPORTATION can alleviate the degradation caused by large $n$ is not obtained by the refined analysis but by its methodology.**
>
> We would like to briefly explain why TELEPORTATION can alleviate this degradation in the following.
>
> The convergence rate of DSGD degrades when $n$ is substantially large since $\mathbf{x}_i$ comes to be far from $\bar{\mathbf{x}}$ due to the sparse communication characteristic.
> To prevent $\mathbf{x}_i$ from being far from $\bar{\mathbf{x}}$, TELEPORTATION activates only $k$ nodes.
> If we use a small $k$, the gossip averaging is performed on a small topology, and the convergence rate does not depend on $p_n$, as shown in Theorem 1.
> Comparing Proposition 1 and Theorem 1, the second and third terms in the convergence rate in Theorem 1 are better since $p_k \geq p_n$, but the first term is worse.
> Finally, by carefully tuning $k$, we can balance these terms, obtaining the statement of Theorem 2, and TELEPORTATION can totally alleviate the degradation caused by large $n$.

---

> ### Author Response · Authors · 2024-11-19
> **Official Comment by Authors (2/2)**
>
> > The proposed hyperparameter-tuning method for selecting the number of active nodes adds another layer of complexity to the method. However, the impact of this additional tuning on overall training time and resource consumption is not fully explored. [...]
>
> We will add the following discussion in the revised manuscript.
>
> By comparing with DSGD, TELEPORTATION has an additional hyperparameter $k$.
> **However, thanks to Algorithm 2, its hyperparameter-tuning requires only $2T$ iteration in total, which is not very expensive.**
>
> From the viewpoint of the convergence rate, TELEPORTATION is superior to DSGD, even considering the cost of hyperparameter-tuning with Algorithm 2 since the cost of hyperparameter-tuning with Algorithm 2 is constant, which is negligible compared to the degradation of the convergence rate of DSGD caused by large $n$.
>
> From an experimental viewpoint, TELEPORTATION can be superior to DSGD in the heterogeneous case.
> Figure 3 shows that TELEPORTATION can train the neural network more stably than DSGD in the heterogeneous case.
> Thus, even considering the cost of hyperparameter-tuning with Algorithm 2, TELEPORTATION can be superior to DSGD.
> In the homogeneous case, DSGD and TELEPORTATION achieved comparable performance in Figure 3.
> Thus, in the homogeneous case, DSGD is superior to TELEPORTATION.
>
>
> > The assumption that any two nodes can communicate directly (as mentioned in the abstract and discussions) is overly idealized. [...]
>
> As the reviewer mentioned, there are cases, such as wireless networks, where there are pairs of nodes that cannot communicate. However, there are also many settings where this condition is satisfied.
> For example, in a data center or in a setting where nodes are connected to the Internet, any two nodes can exchange parameters.
> TELEPORTATION can be used in these cases to scale decentralized learning to a large number of nodes.
>
> Furthermore, we would like to emphasize that even in this setting, it is not trivial to eliminate the convergence rate degradation caused by large $n$.
> In this setting, many existing papers have also tried to alleviate this degradation by designing a topology with a large spectral gap [2-4], while none of the existing papers can completely eliminate this degradation (see the discussion in Section 2.2).
> TELEPORTATION is the first decentralized learning method that can completely alleviate the degradation caused by large $n$.
>
> It is a limitation that TELEPORTATION can only work when any two nodes can communicate.
> However, we believe that our work has achieved the important steps for scaling decentralized learning to a large number of nodes in the general case where there are pairs of nodes that cannot communicate.
>
>
> ## Reference
> [1] Koloskova et. al., A Unified Theory of Decentralized SGD with Changing Topology and Local Updates, In ICML 2019
>
> [2] Takezawa et. al., Beyond Exponential Graph: Communication-efficient Topologies for Decentralized Learning via Finite-time Convergence, In NeurIPS 2023
>
> [3] Ying et. al., Exponential Graph is Provably Efficient for Decentralized Deep Training, In NeurIPS 2021
>
> [4] Ding et. al., . DSGD-CECA: Decentralized SGD with Communication-optimal Exact Consensus Algorithm, In ICML 2023

---

> > ### Comment · Reviewer_bkGn · 2024-11-22
> > **Thanks for the rebuttal.**
> >
> > Thanks for the responses. Most of my questions/concerns have been addressed so I increase my point to 6.
> >
> > Here are some further thoughts.
> >
> > 1. I understand that in the convergence rates, the first term decreases as $n$ increases, while the remaining terms would increase if $n$ is further increased. My point is that not every value of $n$ will decrease the convergence rate. Within a certain range (when the term $1/\sqrt{nT}$ dominates), a larger $n$ might actually accelerate the convergence rate, leading to a linear speedup.
> >
> > 2. I suggest that the authors add a discussion on the lower bound in the revised paper.
> >
> > 3. It would also be beneficial to expand on why TELEPORTATION can alleviate this degradation in the discussion that follows.

---

> > > ### Author Response · Authors · 2024-11-25
> > >
> > > We thank the reviewer for his/her positive feedback.
> > > We promise to include the above discussion in the revised manuscript to make this paper more intuitive for the reader.

---

### Official Review · Reviewer_AkDU · 2024-11-04

**Soundness:** 2
**Presentation:** 4
**Contribution:** 2
**Rating:** 6
**Confidence:** 4

**Summary:**

This paper proposes a novel decentralized algorithm designed to perform well when the number of nodes $n$ is very large. The proposed algorithm synchronizes a smaller set of local states $\\{z_i\\}_{i \in \\{1,...,k\\}}$ for $k < n$ by running a single step of Decentralized SGD (DSGD) [Lian et al., 2017] on a subgraph of $k$ nodes, and then transfer the local states to another set of $k$ nodes for computing the gradients of other local objectives. Such algorithm is suitable for fully-connected distributed system. The proposed theorem suggests that this algorithm enjoys linear speedup by the total number of nodes $n$, at the cost of consuming $k$ local gradient computations only.

**Strengths:**

- The idea of sacrafising gradient updates of the whole network in favor of maintaning lower consensus error is novel.

**Weaknesses:**

- In line 268, $k$ can be simplified as
$$
k = \max\left\\{ 1, \min \left\\{\left\lceil \left(\frac{T(\sigma^2 + \zeta^2)}{Lr_0} \right)^{1/7} \right\rceil, n \right\\} \right\\}
$$
because $a^{1/7} < a^{1/5}$ for any $a > 1$.
- The main claim of Theorem 2 might be theoretically flawed and deserves more attention, which relates to the $n$ linear speedup results. More discussion is provided in the section Questions: (**About the proof of Lemma 9**).

**Questions:**

(**About experiment**)
- I suggest the authors to plot the convergence of consensus error in the experiments as well. This can give a clearer picture to whether the $k$ nodes interaction in the proposed algorithm achieve faster convergence in consensus error than the $n$ nodes interaction in Decentralized SGD.

(**About the proof of Lemma 9**)
- Let's make it clear when does the dominance order changes in (14) for different values of $k$ by the following statements:
$$ \sqrt{A/k} \leq Bk^{2/3} \Leftrightarrow k \ge (A^3 / B^6)^{1/7} \quad \cdots (*)$$
$$ \sqrt{A/k} \leq Ck^{2} \Leftrightarrow k \ge (A / C^2)^{1/5} \quad \cdots (**)$$
Then, by the choice of $k$ in line 1099,
- when $k = \lceil (A^3 / B^6)^{1/7}\rceil \leq \lceil (A / C^2)^{1/5}\rceil$, $(*)$ is true while $(**)$ is false, i.e., $\sqrt{A/k} = \mathcal{O}(Bk^{2/3}) = \mathcal{O}(A^{2/7}B^{3/7})$ and $Ck^2 = C A^{6/7} B^{-12/7}$. Then line 1106 should be $\mathcal{O}(A^{2/7}B^{3/7} + C A^{6/7} B^{-12/7})$ instead.
- when $k = \lceil (A / C^2)^{1/5}\rceil \leq \lceil (A^3 / B^6)^{1/7}\rceil$, $(**)$ is true while $(*)$ is false, so that by the similar arguement line 1112 should be $\mathcal{O}(A^{2/15} B C^{-4/15} + A^{2/5}C^{1/5})$.
- when $k = n \leq \min\{ \lceil (A^3 / B^6)^{1/7}\rceil,  \lceil (A / C^2)^{1/5}\rceil \}$, both $(*)$ and $(**)$ are false and line 1117 should be $\mathcal{O}(\sqrt{A/n} + Bn^{2/3} + Cn^2)$.
Therefore, none of the above cases show a convergence bound that is consistent with the main claim in Theorem 2, i.e., a linear speedup with total number of nodes $n$. May I request the authors to address this issue, and point out my mistakes if I am wrong in the above calculation.
- The similar argument applies to the proof of Lemma 10.

(**Connection to DSGD**)
- We can equivalently interpret the proposed algorithm as a DSGD [Lian et. al., 2017] algorithm of $k$ nodes and each node has access to all local objective $f_i, ~i\in \\{1, ..., n\\}$. Each iteration of the proposed algorithm with an active node set $V^{(t)}$ corresponds to an iteration of the above-mentioned DSGD algorithm where $\\{\nabla f_j(z^{(t)}_{{\rm token\\_id}_j^{(t)}}, \xi_j^{(t)}): v_j \in V^{(t)}\\}$ are the sampled local gradients. Therefore theoretically, by the results of [Lian et. al., 2017], I expect that the proposed algorithm to only achieve the linear speedup of $\mathcal{O}(1/\sqrt{kT})$.
- Also, by the above equivalence, the proposed algorithm is only serving as a new implementation of DSGD on large graph without data heterogeneity, i.e., $\varsigma = 0$ in Assumption 1.3 of [Lian et. al, 2017].

- I suggest the authors to consider along this argument and compare with DSGD in the main text in terms of such equivalence. Also, since this algorithm  is potentially only contributing an efficient large graph implementation of DSGD, I suggest the authors to expand the experiment section with larger scale experiments such as larger dataset and larger number of nodes $n> 100$.

---

> ### Author Response · Authors · 2024-11-22
> **Official Comment by Authors (1/2)**
>
> We thank the reviewer for your constructive feedback.
>
> >  In line 268, $k$ can be simplified as [...]
>
> Thank you for the suggestion.
> We have not revised the statement of our theorems in the main paper yet because revising the theorem would complicate my response to your other comments.
> We will revise it as you suggested in the camera-ready version.
>
> > About the experiments: [...]
>
> We are currently conducting these experiments, but we do not have enough time left to show you the results in this rebuttal period.
> We promise to add these experiments in the camera-ready version.
>
> > About the proof of Lemma 9 [...]
>
> To derive the statement of Lemma 9, further expansion of the equations is necessary.
>
> * When $k = \lceil (A^3 / B^6)^{1/7} \rceil \leq \min \\{ \lceil (A / C^2)^{1/5} \rceil, n \\}$, we get
> \begin{align*}
> \sqrt{\frac{A}{k}} = \mathcal{O} (A^{\frac{2}{7}} B^{\frac{3}{7}}),
> \quad B k^{\frac{2}{3}} = \mathcal{O} (A^{\frac{2}{7}} B^{\frac{3}{7}}),
> \quad C k^2 \leq \mathcal{O} (A^{\frac{2}{5}} C^{\frac{1}{5}}).
> \end{align*}
> In the last inequality, we use $k \leq \lceil (A / C^2)^{1/5} \rceil$. Note that we need to use $k \leq \lceil (A / C^2)^{1/5} \rceil$ instead of $k = \lceil (A^3 / B^6)^{1/7} \rceil$ to get the statement of Lemma 9.
> Then, using the above inequalities, we get the following convergence rate:
> \begin{align*}
>     \mathcal{O} \left( A^{\frac{2}{7}} B^{\frac{3}{7}} + A^{\frac{2}{5}} C^{\frac{1}{5}} \right).
> \end{align*}
>
> * When $k = \lceil (A/C^2)^{1/5} \rceil \leq \min \\{ \lceil (A^3 / B^6)^{1/7} \rceil, n \\}$, we get
> \begin{align*}
> \sqrt{\frac{A}{k}} = \mathcal{O} (A^{\frac{2}{5}} C^{\frac{1}{5}}),
> \quad B k^{\frac{2}{3}} \leq \mathcal{O} (A^{\frac{2}{7}} B^{\frac{3}{7}}),
> \quad C k^2 = \mathcal{O} (A^{\frac{2}{5}} C^{\frac{1}{5}}).
> \end{align*}
> Note that we use $k \leq \lceil (A^3 / B^6)^{1/7} \rceil$ to obtain $B k^{2/3} \leq \mathcal{O} (A^{2/7} B^{3/7})$.
> Then, using the above inequalities, we get the following convergence rate:
> \begin{align*}
>     \mathcal{O} \left( A^{\frac{2}{7}} B^{\frac{3}{7}} + A^{\frac{2}{5}} C^{\frac{1}{5}} \right).
> \end{align*}
>
>
> * When $k=n \leq \min \\{ \lceil (A^3 / B^6)^{1/7} \rceil,  \lceil (A/C^2)^{1/5} \rceil \\}$, we get
> \begin{align*}
> \sqrt{\frac{L r_0 (\sigma^2 + (1 - \frac{k-1}{n-1})\zeta^2)}{k T}} = \sqrt{\frac{L r_0 \sigma^2}{n T}},
> \quad B k^{\frac{2}{3}} \leq \mathcal{O} (A^{\frac{2}{7}} B^{\frac{3}{7}}),
> \quad C k^2 \leq \mathcal{O} (A^{\frac{2}{5}} C^{\frac{1}{5}}),
> \end{align*}
> where we use $k \leq \lceil (A^3 / B^6)^{1/7} \rceil$ and $k \leq \lceil (A/C^2)^{1/5} \rceil$.
> Then, we get the following rate:
> \begin{align*}
>     \mathcal{O} \left(\sqrt{\frac{L r_0 \sigma^2}{n T}} + A^{\frac{2}{7}} B^{\frac{3}{7}} + A^{\frac{2}{5}} C^{\frac{1}{5}} \right).
> \end{align*}
>
>
> By summarizing the above inequalities, we obtain the statement of Lemma 9.
> We have clarified the explanation above in the revised manuscript, with the revisions highlighted in blue.
>
>
>
> > [...] I expect that the proposed algorithm to only achieve the linear speedup of $\mathcal{O}(1 / \sqrt{kT})$.
>
> If we carefully tune $k$, the convergence rate of TELEPORTATION can be $\mathcal{O}(\frac{1}{\sqrt{n T}})$ as shown in Theorem 2.
>
> Here, we intuitively explain why setting $k$ as in Theorem 2 achieves the linear speedup $\mathcal{O}(\frac{1}{\sqrt{n T}})$.
> See the proof in Lemma 9 and the above response for the precise discussion.
>
> * When $T$ is small, the second and third terms $\mathcal{O}((\frac{(1- p_k)}{T^2 p_k})^{1/3} + \frac{1}{T p_k})$ dominate the convergence rate in Theorem 1.
> In this case, we would like to use a small $k$.
> * When $T$ is large, the first term $\mathcal{O}(\frac{1}{\sqrt{k T}})$ dominates the convergence rate in Theorem 1.
> In this case, we would like to set $k=n$.
>
> Thus, if we carefully set $k$ so that $k$ increases to $n$ as $T$ increases, we can balance $\mathcal{O}(\frac{1}{\sqrt{k T}})$ and $\mathcal{O}((\frac{(1- p_k)}{T^2 p_k})^{1/3} + \frac{1}{T p_k})$. Then, TELEPORTATION can achieve the linear speedup $\mathcal{O}(\frac{1}{\sqrt{nT}})$ and can totally eliminate the degradation caused by large $n$.

---

> ### Author Response · Authors · 2024-11-22
> **Official Comment by Authors (2/2)**
>
> > Also, by the above equivalence, the proposed algorithm is only serving as a new implementation of DSGD on large graph without data heterogeneity [...]
>
> We respectfully disagree with this reviewer's comment that TELEPORTATION is equivalent to DSGD in the homogeneous case. Even in the homogeneous case, DSGD suffers from a degradation in convergence rate when $n$ is substantially large because stochastic gradient noise causes the parameters held by each node to drift away.
> Activating only an appropriate number of nodes in TELEPORTATION is critical to mitigating this issue in both homogeneous and heterogeneous cases.
>
> Specifically, TELEPORTATION with ring and DSGD achieve the following convergence rate when $\zeta=0$, respectively:
> \begin{align*}
>     &\textbf{TELEPORTATION}: \\;\\; \mathcal{O} \left( \sqrt{\frac{L r_0 \sigma^2}{n T}} + \left( \frac{L r_0 \sigma^{\frac{3}{2}}}{T }\right)^\frac{4}{7} + \left( \frac{L r_0 \sigma^\frac{4}{3}}{T} \right)^\frac{3}{5} + \frac{L r_0}{T} \right), \\\\
>     &\textbf{DSGD}: \\;\\; \mathcal{O} \left( \sqrt{\frac{L r_0 \sigma^2}{n T}} + \left( \frac{L^2 r_0^2 \sigma^2 (1 - p_n)}{T^2 p_n} \right)^\frac{1}{3} + \frac{L r_0}{T p_n} \right).
> \end{align*}
> The convergence rate of DSGD degrades when $n$ is substantially large, while the convergence rate of TELEPORTATION is consistently improved as $n$ increases.
> Our experiments in Figure 2 can also show the superiority of TELEPORTATION, demonstrating that TELEPORTATION can converge faster than DSGD in both homogeneous and heterogeneous cases.
>
> > I suggest the authors to consider along this argument and compare with DSGD in the main text in terms of such equivalence.
>
> Thank you for your suggestion.
> We will clarify this point in the revised manuscript.
>
> > I suggest the authors to expand the experiment section with larger scale experiments such as larger dataset and larger number of nodes $n > 100$.
>
> Since there is not enough time left until the end of the rebuttal, we cannot show the results with a larger number of nodes in this rebuttal period. We will add the results with a larger number of nodes in the camera-ready version.

---

> > ### Comment · Reviewer_AkDU · 2024-11-22
> > **Response to Authors Comment**
> >
> > Thank you the authors for the clear response. I am convinced by the response and will raise the rating to 6.
> > I suggest the authors to include the above discussion around different regime of $T$ (e.g., $T \leq n^7 \Rightarrow k = \mathcal{O}(T^{1/7})$ vs $T > n^7 \Rightarrow k = n$ for ring graph) and the corresponding choices of $k$ in the main text after the paragraph of Theorem 2, so that readers can make a clear interpretation of the results.

---

> ### Author Response · Authors · 2024-11-22
>
> We thank the reviewer again for carefully reviewing our theorem and response.
> We promise to include our discussion in the camera-ready version and revise the paper to make our methods more intuitive to the reader.

---

### Official Review · Reviewer_Neby · 2024-11-06

**Soundness:** 3
**Presentation:** 3
**Contribution:** 3
**Rating:** 6
**Confidence:** 3

**Summary:**

This work studies a decentralized optimization algorithm whose convergence rate does not deteriorate with increasing number of nodes. Usually, when number of nodes in a network increases, the spectral gap, i.e., $(1 - p_n)$ (according to the notation of the paper), also increases. Larger spectral gap means that decentralized SGD requires a larger number of iterations to converge.

In order to make decentralized SGD scalable for large networks, the proposed algorithm randomly activates a small subset of nodes by randomly sampling them. These nodes get the updated models from nodes activated in the last iteration, does a descent step, and subsequently communicate with other nodes active in the current iteration to do a local consensus step (only amongst the currently activated node). this process is repeated over multiple iterations.

The paper analyzes the convergence of this algorithm, and also provides an efficient algorithm to tune the number of active nodes.

**Strengths:**

The paper is generally well-written. The problem statement being addressed is clear, and the proposed algorithm is also quite simple. The paper does a commendable job analyzing their proposed algorithm, in addition to validating numerically. The assumptions are also clearly stated, which are quite standard in decentralized optimization.

The work also compares their algorithm with client sampling, which is a natural question that came to my mind while reading the paper, and I appreciate that it was answered. I have a few concerns and would be glad if these are addressed. Despite my concerns, I strongly believe that within the current scope of the problem statement considered in the paper, it is adequately addressed.

**Weaknesses:**

I would appreciate it if the authors would answer some of my concerns.

1. How are the consensus weights $\mathbf{W}$ chosen, and how is it assumed that the randomly activated nodes at any iteration know these weights?

2. How are the nodes sampled? Are they sampled uniformly at random? Would it help if the nodes were sampled taking some other criteria in mind, such as activating nodes that are more likely to have an edge between them? More generally, how do the authors expect the sampling to be implemented in practice in a fully decentralized topology? More generally, can the authors highlight a practical application scenario?

3. The paper would benefit with some explicit discussion on the dependence of the node-selection strategy, and/or the required number of active nodes on the network topology? For instance, it seems like if the network is sparse, perhaps more nodes need to be activated at every iteration?

4. Will it make sense to change the number of activated nodes in every iteration?

I must acknowledge that I am not completely familiar with the current literature in topology selection for decentralized optimization. So I cannot definitively comment on the novelty of the paper, but based on how it is placed in the writing itself, the contribution seems non-trivial (despite the weaknesses mentioned here).

**Questions:**

I have some more questions which are not very critical, but I believe that the paper will benefit from:

1. In the current proposed algorithm, data heterogeneity is not taken into account. Some discussions on how the algorithm might need to be modified in the presence of data heterogeneity would be appreciated. Perhaps optimizing the weight matrix $\mathbf{W}$ would be helpful?

2. Although less related, in light of the popularity of federated learning literature, the privacy of activated nodes might be an important criteria to consider. Since each node knows which node participated in the previous iteration, this can lead to some degree of privacy leakage, that needs to be quantified. (Please note that I do not see this as a major drawback of the paper -- it is just a suggestion). It ties more with my previous question of practical application scenario, where the network has a large number of nodes, so privacy becomes important.

**Details Of Ethics Concerns:**

None needed.

---

> ### Author Response · Authors · 2024-11-20
> **Official Comment by Authors (1/2)**
>
> We thank the reviewer for your positive feedback.
>
> > How are the consensus weights $\mathbf{W}$ chosen, and how is it assumed that the randomly activated nodes at any iteration know these weights?
>
> We explain how the sampling of active nodes can be implemented in more detail.
>
> All nodes communicate before starting the training and have the same seed value, as we mentioned in line 215.
> Then, we sample the active nodes $V_\text{active}$ and assign $\\{1, 2, \cdots, k \\}$ to variables $\\{ \\text{token\\_id}_i \\}$ by using this seed value for each iteration in line 3 in Algorithm 1.
> Since all nodes have the same seed value, all nodes can know which nodes are active and can obtain the same variables $\\{ \text{token\\_id}_i \\}$ without communicating with other nodes.
> Using these variables, active nodes exchange parameters with their neighbors and compute the weighted average in lines 11-12 in Algorithm 1.
>
> > Would it help if the nodes were sampled taking some other criteria in mind, such as activating nodes that are more likely to have an edge between them?
>
> In our current proposed method, we assume that the active nodes are sampled uniformly.
> In the federated learning literature, many client sampling strategies have been proposed, as we summarized in Sec. 4.
> Thus, there might be a better scheme for selecting active nodes than random sampling.
> However, the primary objective of our work is to address the convergence rate degradation associated with large $n$ and to make decentralized learning scalable to a large number of nodes.
> We have shown that random sampling of active nodes successfully achieves this goal.
> We believe that proposing a better node sampling scheme is beyond the scope of our paper, and we leave it for future research.
>
>
> > More generally, how do the authors expect the sampling to be implemented in practice in a fully decentralized topology? More generally, can the authors highlight a practical application scenario?
>
> Sampling of the active nodes can be performed in a decentralized manner.
> See our first response for more details.
>
> > The paper would benefit with some explicit discussion on the dependence of the node-selection strategy, and/or the required number of active nodes on the network topology? For instance, it seems like if the network is sparse, perhaps more nodes need to be activated at every iteration?
>
> In Theorem 2, we show the optimal number of active nodes when we use a ring and exponential graph as a topology to connect active nodes.
> An exponential graph has a larger spectral gap than a ring.
> Comparing the results with a ring and an exponential graph, more nodes are activated when we use an exponential graph.
> This is because the parameters that each node has are more likely to drift away when the number of active nodes increases when a ring is used than when an exponential graph is used.
>
> See the above response for the discussion of the node-selection strategy.

---

> > ### Author Response · Authors · 2024-11-20
> > **Official Comment by Authors (2/2)**
> >
> > > Will it make sense to change the number of activated nodes in every iteration?
> >
> > To simplify our analysis, we considered the setting where the number of active nodes is constant.
> > If we consider a setting where the number of active nodes can change in every iteration, we might further improve the convergence rate of TELEPORTATION.
> > We believe that it is one of the most promising directions for future research.
> >
> > > In the current proposed algorithm, data heterogeneity is not taken into account. Some discussions on how the algorithm might need to be modified in the presence of data heterogeneity would be appreciated. [...]
> >
> > Many existing papers tried to make DSGD robust to data heterogeneity by using variance reduction,
> > proposing decentralized learning methods whose convergence rates are independent of data heterogeneity $\zeta$ [1,2,3].
> > Similarly, it may be possible to make the convergence rate of TELEPORTATION independent of data heterogeneity by leveraging variance reduction.
> > However, we believe that making TELEPORTATION robust to data heterogeneity is out of scope and leave it for future research.
> >
> >
> > > Although less related, in light of the popularity of federated learning literature, the privacy of activated nodes might be an important criteria to consider. Since each node knows which node participated in the previous iteration, this can lead to some degree of privacy leakage, that needs to be quantified. [...]
> >
> > Differential privacy is the most common technique for ensuring privacy in federated learning literature.
> > In the following, we briefly discuss the noise to be added to implement differential privacy.
> >
> > As the reviewer pointed out, TELEPORTATION may necessitate adding substantial noise since only a few nodes are activated.
> > However, since TELEPORTATION can converge faster than DSGD, it may allow us to reduce the required noise.
> > Differential privacy requires significant noise as the number of iterations increases [4].
> > Therefore, it is not trivial to draw a conclusion here, and we would like to leave it for future research.
> >
> >
> > ## Reference
> > [1] Pu et. al., Distributed stochastic gradient tracking methods, In Mathematical Programming 2021
> >
> > [2] Yuan et. al., Exact diffusion for distributed optimization and learning—Part I: Algorithm development, In IEEE Transactions on Signal Processing 2018
> >
> > [3] Tang et. al., $D^ 2$: Decentralized training over decentralized data, In International Conference on Machine Learning 2018
> >
> > [4] Abadi et. al., Deep learning with differential privacy, In ACM Conference on Computer and Communications Security 2016

---

> > > ### Comment · Reviewer_Neby · 2024-12-01
> > > **Ack**
> > >
> > > Thank you for your response.
> > >
> > > I will be keeping my score, as I believe this paper is good and leaning towards acceptance.
> > >
> > > Justification for not higher score: Lots of questions as discussed above are open, and there is a huge score for improvement. I'd increase my score to 7 if ICLR provisioned that.

---

### Meta-Review · Area_Chair_BA24 · 2024-12-26

**Metareview:**

This paper presents TELEPORTATION, a decentralized optimization algorithm designed to mitigate the common issue of deteriorating convergence rates in decentralized SGD as the number of nodes increases. The key insight is that larger spectral gaps in the communication graph typically require more iterations for convergence. TELEPORTATION addresses this by randomly activating a small subset of nodes at each iteration, reducing communication costs while maintaining effective learning. These active nodes update the model, perform a descent step, and achieve local consensus with other active nodes, repeating this process over multiple iterations. The paper provides a theoretical analysis of the algorithm’s convergence and an efficient method for tuning the number of active nodes.

There were a few concern raised by reviewers that some of them are partially resolved by authors, but the overall consensus was on accepting the paper; noting that the paper needs a through revision to incorporate suggested modifications.

**Additional Comments On Reviewer Discussion:**

Presentation of claimed results needs further clarification (please consult suggestions from reviewers)

---

### Decision · Program_Chairs · 2025-01-22

Accept (Poster)